# Flexible Tails for Normalizing Flows

**Tennessee Hickling** [* 1]   **Dennis Prangle** [* 1]

## Abstract

Normalizing flows are a flexible class of probability distributions, expressed as transformations of a simple base distribution. A limitation of standard normalizing flows is representing distributions with heavy tails, which arise in applications to both density estimation and variational inference. A popular current solution to this problem is to use a heavy tailed base distribution. We argue this can lead to poor performance due to the difficulty of optimising neural networks, such as normalizing flows, under heavy tailed input. We propose an alternative, "tail transform flow" (TTF), which uses a Gaussian base distribution and a final transformation layer which can produce heavy tails. Experimental results show this approach outperforms current methods, especially when the target distribution has large dimension or tail weight.

## 1. Introduction

A normalizing flow (NF) expresses a complex probability distribution as a parameterised transformation of a simpler base distribution. A NF sample is

$$x = T(z; \theta), \qquad (1)$$

where $z$ is a sample from the base distribution, typically $\mathcal{N}(0, I)$. A number of transformations have been proposed which produce flexible and tractable distributions. Typically multiple transformations are composed to form $T$ with the desired level of flexibility. Applications include density estimation (fitting a transformation to observed data points) and variational inference (fitting a transformation to a target distribution). In either case $\theta$, the parameters controlling $T$, can be optimised using stochastic gradient methods for a suitable objective function. For reviews of NFs see Kobyzev et al. (2020) and Papamakarios et al. (2021).

---

[*]Equal contribution   [1]School of Mathematics, University of Bristol, Bristol, UK. Correspondence to: Tennessee Hickling <tennessee.hickling@bristol.ac.uk>.

*Proceedings of the $42^{nd}$ International Conference on Machine Learning*, Vancouver, Canada. PMLR 267, 2025. Copyright 2025 by the author(s).

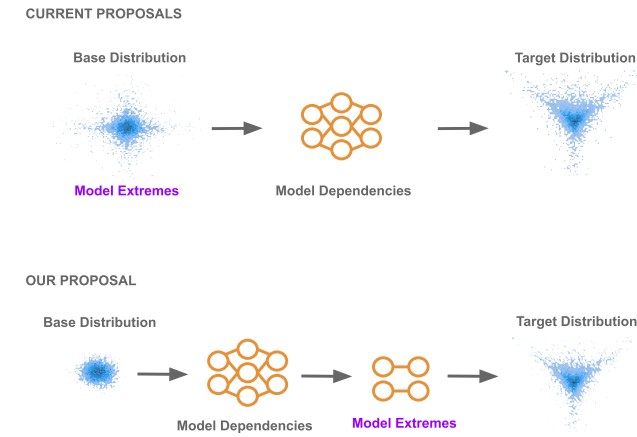

Figure 1: Our method models extremes in the final layer, not the base distribution.

Modelling distributions with heavy tails is critical in many applications such as climate (Zscheischler et al., 2018), contagious diseases (Cirillo & Taleb, 2020) and finance (Gilli & Këllezi, 2006). Indeed, in the context of density estimation, results from extreme value theory (EVT) (Coles, 2001; Embrechts et al., 2013) show distributional tails can often be modelled using a particular heavy tailed distribution, the generalized Pareto distribution (GPD). Heavy tailed targets can also arise naturally in Bayesian inference, requiring heavy tailed approximate distributions in variational inference (Liang et al., 2022) —and providing challenges for MCMC methods (see e.g. Yang et al., 2024).

However standard NFs do not model heavy tails well. In particular, Jaini et al. (2020) prove that Gaussian tails cannot be mapped to GPD tails under Lipschitz transformations. (This is a special case of their main result, which we include as Theorem 1.2 below.) Many NFs use Lipschitz transformations of Gaussian base distributions, implying they produce distributions poorly approximating heavy tailed distributions.

Based on this result, several authors have proposed using heavy tailed base distributions. This includes the **tail adaptive flow** (TAF) methods of Laszkiewicz et al. (2022) for density estimation, and similar methods for variational inference (Liang et al., 2022). We argue that using heavy tailed

base distributions has an important drawback. Typical NF transformations involve passing the input $z$ through a neural network. However neural network optimisation can perform poorly under heavy tailed input—as this can induce heavy tailed gradients, which are known to be problematic (Zhang et al., 2020). We verify this problem empirically in Section 4.1 for NFs (and in Appendix F for a simpler neural network regression example.)

To improve performance we propose an alternative: use a Gaussian base distribution with a final non-Lipschitz transformation in $T$. We refer to this as the **tail transform flow** (TTF) approach. The difference is illustrated in Figure 1. In Section 3.1 we provide a suitable final transformation (3), motivated by extreme value theory. This is easy to implement in automatic differentiation libraries as it's based on a standard special function: the complementary error function. We also prove that this transformation converts Gaussian tails to heavy tails with tunable tail weights, and hence provides the capability of producing heavy tails when used with standard flows.

We perform experiments showing our TTF approach outperforms current methods, especially when the target distribution has large dimension or tail weight. We concentrate on density estimation, investigating both synthetic and real data, but also include a small-scale variational inference experiment.

To summarise, our contributions are:

- We illustrate the problem of using extreme inputs in NFs.

- We introduce the TTF transformation—equation (3)—and prove it converts Gaussian tails to heavy tails with tunable tail weights. This provides the capability of producing heavy tails when used with standard NFs.

- We provide practical methodology using TTF in a normalizing flow architecture.

- We demonstrate improved empirical results for density estimation (synthetic and real data examples) and variational inference (an artificial target example) compared to standard NFs, and other NF methods for heavy tails.

In the remainder of this section we review related prior work. Then Section 2 outlines background material and Section 3 describes our proposed transformation, as well as summarising our theoretical results. Section 4 contains our examples, and Section 5 concludes. The appendices contain further technical details, including proofs. Code for all our examples can be found at `https://github.com/Tennessee-Wallaceh/tailnflows`.

## 1.1. Related Work

### 1.1.1. THEORY

A common definition of heavy tails (e.g. Foss et al., 2011) is as follows.

**Definition 1.1.** A real valued random variable $Z$ is (right-) **heavy-tailed** when $\mathbb{E}[\exp(\lambda Z)] = \infty$ for all $\lambda > 0$. Otherwise $Z$ is **light-tailed**.

Examples of heavy tailed distributions include Student's $T$ and Pareto distributions. An example of a light tailed distribution is Gaussian distributions.

Under Definition 1.1 the following result holds.

**Theorem 1.2** (Jaini et al., 2020). *Let $Z$ be a light tailed real valued random variable and $T$ be a Lipschitz transformation. Then $T(Z)$ is also light tailed.*

Jaini et al. (2020) also generalise this result to the multivariate case, and prove that several popular classes of NFs use Lipschitz transformations (see also Liang et al., 2022).

Other mathematical definitions of tail behaviour exist, including alternative usages of the term "heavy tails". Liang et al. (2022) apply such definitions to prove more detailed results on how Lipschitz transformations affect tail weights (e.g. Theorem C.7 in our appendices).

To model heavy tailed distributions, Jaini et al. note that there is a choice "of either using source densities with the same heaviness as the target, or deploying more expressive transformations than Lipschitz functions". They pursue the former approach and this paper investigates the latter.

### 1.1.2. HEAVY TAILED BASE DISTRIBUTIONS

Several NF papers propose using a heavy tailed base distribution. To discuss these, let $Z$ be a random vector following the base distribution of the normalizing flow, and let $Z_i$ be its $i$th component.

Firstly, for density estimation, Jaini et al. (2020) set $Z_i \sim t_\nu$: a standard (location zero, scale one) Student's T distribution with $\nu$ degrees of freedom. The $Z_i$ components are independent and identically distributed. The $\nu$ parameter is learned jointly with the NF parameters. Optimising their objective function requires evaluating the Student's $T$ log density and its derivative with respect to $\nu$, which is straightforward.

An extension is to allow **tail anisotropy**—tail behaviour varying across dimensions—by setting (independently) $Z_i \sim t_{\nu_i}$, so the degrees of freedom can differ with $i$. In Laszkiewicz et al. (2022), two such approaches are proposed for density estimation. Marginal Tail-Adaptive Flows (mTAF) first learn $\nu_i$ values using a standard estimator (a version of the Hill, 1975 estimator). Then the distribution of $Z$ is fixed while learning $T$. Another feature of mTAF

is it allows Gaussian marginals by using Gaussian base distributions for appropriate $Z_i$ components, and then keeping Gaussian and non-Gaussian components separate throughout the flow transformation. Generalised Tail-Adaptive Flows (gTAF) train the $\nu_i$s and $\theta$ jointly. Liang et al. (2022) propose a similar method to gTAF for the setting of variational inference: Anisotropic Tail-Adaptive Flows (ATAF).

### 1.1.3. OTHER METHODS

Here we discuss work on alternatives to a Student's T base distribution. In the setting of density estimation, Amiri et al. (2022) consider two alternative base distributions: Gaussian mixtures, and generalized Gaussian. Although the former are light tailed (see Appendix A), they argue that mixture distributions can in theory model any smooth density given enough components, and also that they are more stable in optimisation than heavy tailed base distributions. The latter has tails which can be heavier than Gaussian, but are lighter than Student's T.

McDonald et al. (2022) propose COMET (copula multivariate extreme) flows for the setting of density estimation. This involves a preliminary stage of estimating marginal cumulative distribution functions (cdfs) for each component of the data. The body of each marginal is approximated using kernel density estimation, and the tails are taken as GPDs, with shape parameters chosen using maximum likelihood. The COMET flow then uses a Gaussian base distribution, followed by a **copula transformation**. This is an arbitrary normalizing flow followed by an elementwise logistic transformation so that output is in $[0, 1]^d$. Finally the inverse marginal cdfs are applied to each component. In Appendix C.2 we analyse the resulting tail behaviour and prove that the output cannot capture the full range of heavy tailed behaviour. More formally, Theorem C.8 proves that the output is not in the Fréchet domain of attraction—see Definition 2.1 below. This is in contrast to our proposed method, TTF, which we show in Appendix C.3 does produce in outputs in the Fréchet domain. Despite this our experiments in Section 4 find COMET flows often have good empirical performance, which we discuss in Section 5.

The extreme value theory literature also proposes methods to jointly model heavy tails and the body of a distribution. See Huser et al. (2025) (Section 4) for a review of several approaches. We comment on one particularly relevant univariate method: Papastathopoulos & Tawn (2013); Naveau et al. (2016); de Carvalho et al. (2022) use a cdf $G \circ H(y)$ which composes the GPD cdf $H$ with a **carrier function** $G : [0, 1] \rightarrow [0, 1]$ which is taken to be a simple parametric function. Similarly, Stein (2021) composes several carrier functions. Our approach is similar, effectively using NFs in place of carrier functions, and covering the multivariate case. Naveau et al. (2016) proves that certain conditions

on the carrier function provide useful theoretical properties, and it would be interesting to explore analogous results for our method in future.

## 2. Background

### 2.1. Normalizing Flows

Consider vectors $z \in \mathbb{R}^d$ and $x = T(z) \in \mathbb{R}^d$. Suppose $z$ is a sample from a base density $q_z(z)$. Then the transformation $T$ defines a **normalizing flow** density $q_x(x)$. Suppose $T$ is a diffeomorphism (a bijection where $T$ and $T^{-1}$ are differentiable), then the standard change of variables formula gives

$$q_x(x) = q_z(T^{-1}(x))|\det J_{T^{-1}}(x)|. \tag{2}$$

Here $J_{T^{-1}}(x)$ denotes the Jacobian of the inverse transformation and $\det$ denotes determinant.

Typically a parametric transformation $T(z; \theta)$ is used, and $q_z(z)$ is a fixed density, such as that of a $\mathcal{N}(0, I)$ distribution. However some previous work uses a parametric base density $q_z(z; \theta)$. (So $\theta$ denotes parameters defining both $T$ and $q_z$.) For instance some methods from Section 1.1 use a Student's $T$ base distribution with variable degrees of freedom.

Usually we have $T = T_K \circ \ldots \circ T_2 \circ T_1$, a composition of several simpler transformations. Many such transformations have been proposed, with several desirable properties. These include producing flexible transformations and allowing evaluation (and differentiation) of $T$, $T^{-1}$, and the Jacobian determinant. Such properties permit tractable sampling via (1), and density evaluation via (2).

Throughout the paper we describe NFs in the **generative direction**: by defining the transformations $T_i$ applied to a sample from the base distribution. NFs can equivalently be described in the **normalizing direction** by defining the $T_i^{-1}$ transformations. We pick the generative direction simply to be concrete.

### 2.2. Density Estimation

Density estimation aims to approximate a target density $p(x)$ from which we have independent samples $\{x_i\}_{i=1}^N$. We assume $x_i \in \mathbb{R}^d$.

We can fit a normalizing flow by minimising the objective

$$\mathcal{J}_{\text{DE}}(\theta) = -\sum_{i=1}^N \log q_x(x_i; \theta).$$

This is equivalent to minimising a Monte Carlo approximation of the Kullback-Leibler divergence $KL[p(x)||q_x(x; \theta)]$. The objective gradient can be numerically evaluated using automatic differentiation. Thus optimisation is possible by stochastic gradient methods. This approach remains feasible

under a Student's $T$ base distribution since its log density and required derivatives are tractable.

## 2.3. Variational Inference

Variational inference (VI) aims to approximate a target density $p(x)$, often a Bayesian posterior for parameters $x \in \mathbb{R}^d$. Typically VI is used where only an unnormalised target $\tilde{p}(x)$ can be evaluated. Then $p(x) = \tilde{p}(x)/\mathcal{Z}$ but the normalizing constant $\mathcal{Z} = \int_{\mathbb{R}^d} \tilde{p}(x)dx$ cannot easily be calculated.

VI aims to minimise $KL[q_x(x;\theta)||p(x)]$ over a parameterised set of densities $q_x(x;\theta)$. In this paper $q_x(x;\theta)$ is a normalizing flow. An equivalent optimisation task is maximising the **ELBO** objective

$$\mathcal{J}_{\text{VI}}(\theta) = \mathbb{E}_{x \sim q_x}[\log \tilde{p}(x) - \log q_x(x;\theta)].$$

This has a tractable unbiased gradient estimate

$$M^{-1} \sum_{i=1}^{M} \nabla_\theta [\log \tilde{p}(x_i) - \log q_x(x_i;\theta)],$$

where $x_i = T(z_i; \theta)$ and $\{z_i\}_{i=1}^{M}$ are independent samples from the base distribution. Again, the gradient estimate can be numerically evaluated using automatic differentiation, allowing optimisation by stochastic gradient methods.

A generalisation of the above is to use a base distribution with parameters. In particular, ATAF (Liang et al., 2022) needs to learn degrees of freedom for Student's $T$ distributions. The above approach remains feasible by using a Student's $T$ sampling scheme which allows application of the reparameterization trick, as detailed in Appendix G.4.

For more background on this form of VI see Rezende & Mohamed (2015); Blei et al. (2017); Murphy (2023).

## 2.4. Extreme Value Theory

Extreme value theory (EVT) is the branch of statistics studying extreme events (Coles, 2001; Embrechts et al., 2013). A classic result is **Pickands theorem** (Pickands III, 1975); see Papastathopoulos & Tawn (2013) for a review. Given a scalar real-valued random variable $X$, consider the scaled excess random variable $\frac{X-u}{h(u)}|X > u$, where $u > 0$ is a large threshold and $h(u) > 0$ is an appropriate scaling function. The theorem states that if the scaled excess converges in distribution to a non-degenerate distribution, then it converges to a Generalized Pareto distribution (GPD).

A common EVT approach is to fix some $u$, treat $h(u)$ as constant, and model tails of distributions as having GPD densities. The motivation is that this should be a good tail approximation near to $u$, while for $x \gg u$ there is not enough data to estimate the behaviour of $h(u)$.

The GPD distribution involves a shape parameter, $\lambda \in \mathbb{R}$. For $\lambda > 0$, the GPD density is asymptotically (for large $x$) proportional to $x^{-1/\lambda-1}$, while for $\lambda < 0$ the upper tail has bounded support. In terms of Definition 1.1, $\lambda > 0$ guarantees heavy tails and $\lambda < 0$ guarantees light tails. Given $X$, the shape parameter of the GPD resulting from Pickands theorem is a measure of how heavy the tail of $X$ is. A Gaussian distribution results in $\lambda = 0$ (and requires a non-constant scaling function $h(u)$), and $\lambda > 0$ represents heavier tails. Finally, we will use the following definition in our theoretical results later.

**Definition 2.1.** The **Fréchet domain of attraction** with shape parameter $\lambda$, $\Theta_\lambda$, is the set of distributions resulting in $\lambda > 0$ under Pickands theorem.

# 3. Methods and Theory

We propose producing normalizing flow samples $R \circ T_{\text{body}}(z)$, where $z$ is a $\mathcal{N}(0,1)$ sample, $T_{\text{body}}$ is a standard normalizing flow transformation and $R$ is a final transformation. If $T_{\text{body}}$ is a Lipschitz transformation, then the input to $R$ has light tails by Theorem 1.2. Thus the resulting architecture avoids the problem outlined in Section 1 of passing extreme values as inputs to any neural network layers. The final transformation $R$ should be able to output heavy tails of any desired tail weight. This section presents our proposal for a suitable transformation $R$.

Section 3.1 describes our proposed transformation for the univariate case. Section 3.2 summarises our theoretical results on its tail behaviour. Sections 3.3—3.4 give further details to produce a practical general-purpose method, and Section 3.5 comments on universality. Proofs and technical details are given in the appendices. This also includes a discussion of alternative transformations and related results in the literature, in Appendix D.

## 3.1. TTF Transformation

We propose the **tail transform flow** (TTF) transformation $R : \mathbb{R} \to \mathbb{R}$,

$$R(z; \lambda_+, \lambda_-) = \mu + \sigma \frac{s}{\lambda_s} [\text{erfc}(|z|/\sqrt{2})^{-\lambda_s} - 1]. \quad (3)$$

We use the notation $s = \text{sign}(z)$, with $\lambda_s = \lambda_+$ for $s = 1$ and $\lambda_s = \lambda_-$ for $s = -1$. The transformation is based on erfc, the **complementary error function**. This is a special function, reviewed in Appendix B.1, which is tractable for use in automatic differentiation using standard libraries. The parameters $\lambda_+ > 0, \lambda_- > 0$ control tail weights for the upper and lower tails (also often referred to as right and left tails respectively). This allows us to model asymmetry in tail behaviour. The parameters $\mu \in \mathbb{R}$ and $\sigma > 0$ are location and scale parameters. We found these helped performance, although similar effects could be achieved in principle by adjusting $T_{\text{body}}$.

To perform density evaluation via (2) we need to evaluate

the inverse and derivative of (3). These, and some other properties, are provided in Appendix B.3.

## 3.2. Theory

Appendix C.3 proves results on the asymptotic properties of (3). Informally, if $X$ has Gaussian tails then $R(X)$ is in the Fréchet domain of attraction (see Definition 2.1) with $\lambda_+, \lambda_-$ controlling the tail shape parameters. A special case is that $\mathcal{N}(0, 1)$ tails produce GPD output with shape parameters $\lambda_+, \lambda_-$.

So composing $R$ with existing NFs permits the output to have heavy tails with parameterised weights. This is shown by the following argument. Most NFs use a Gaussian base distribution and a Lipschitz transformation. By Theorem 1.2 the output has light tails. Since the NF transformation can be the identity, it is capable of producing Gaussian tails.

## 3.3. Multivariate Transformation

To extend our univariate transformation $R : \mathbb{R} \to \mathbb{R}$ to the multivariate case, we simply transform each marginal with its own $\mu, \sigma, \lambda_+, \lambda_-$ parameters. In some cases we know particular marginals are light tailed. Then we could simply perform an identity transformation instead. However, we find fixing $\lambda_+, \lambda_-$ to low values (we use $1/1000$) suffices.

A more flexible approach would be to allow dependence, for instance by using an autoregressive structure (Papamakarios et al., 2021) to generate the $\lambda_+, \lambda_-$ parameters for each marginal. This could capture tail behaviour that varies in different parts of the distribution. However, exploratory work found that this approach is harder to optimise, so we leave it for future research.

## 3.4. Two Stage Procedure

Joint optimisation of $R$ and $T_{\text{body}}$ can require careful initialisation of the $\lambda$ parameters. (Details of how we do so are provided in Appendix G.3.) As an alternative, we propose a two stage procedure for density estimation: **TTFfix**. Here the tail weight $\lambda_-, \lambda_+$ parameters of $R$ are estimated in an initial step and then fixed while optimising $T_{\text{body}}$ (and the $\mu, \sigma$ parameters of $R$.) This can be viewed as first transforming the data using $R$ to remove heavy tails, and then fitting a standard normalizing flow to the transformed data. Similar approaches appear in McDonald et al. (2022) and Laszkiewicz et al. (2022).

Shape parameter estimators exist in the EVT literature, which we can apply to each marginal tail. We follow Laszkiewicz et al. (2022) in using the Hill double-bootstrap estimator (Danielsson et al., 2001; Qi, 2008). Note that this enforces $\lambda_- = \lambda_+$. Alternatively McDonald et al. (2022) perform maximum likelihood estimation on the highest and lowest 5% of data, to produce tail parameters for the positive and negative tails respectively.

We do not consider a similar two stage procedure for VI, as preliminary estimation of tail weights from the unnormalised target distribution is not straightforward. However recent work on static analysis of probabilistic programs (Liang et al., 2024) provides progress in this direction.

## 3.5. Universality

It's been proved that some NFs have a universality property: "the flow can learn any target density to any required precision given sufficient capacity and data" (Kobyzev et al., 2020). In Appendix E we show that many NF universality results are preserved when the TTF transformation is added as a final layer.

As we've already seen, the situation under bounded capacity is different. Standard NFs cannot produce heavy tailed distributions (Jaini et al. 2020, reviewed as Theorem 1.2 above). However adding our transformation does permit these (see Section 3.2).

So theoretically our method improves the set of distributions which can be modelled under bounded capacity without sacrificing expressiveness in the limit of infinite capacity. The next section shows this is reflected by improved empirical performance modelling heavy tailed data.

# 4. Experiments

This section contains our experiments. Firstly, recall that one motivation for our work in Section 1 is the claim that neural network optimisation can perform poorly under heavy tailed inputs. We verify this empirically in Section 4.1 for NFs. This is also verified in Appendix F for a simple neural network regression example. Secondly, Sections 4.1—4.3 contain normalizing flow examples, comparing our method to existing approaches for density estimation and a proof-of-concept variational inference example. Additional implementation details are provided in Appendix G.

## 4.1. Density Estimation with Synthetic Data

This experiment looks at density estimation for data generated from the following model, with $d > 1$:

$$\{X_i\}_{i=1}^{d-1} \sim t_\nu, \quad X_d | X_{d-1} \sim \mathcal{N}(X_{d-1}, 1). \quad (4)$$

In this model the only non-trivial dependency to learn is between $X_{d-1}$ and $X_d$, but there are also several heavy tailed nuisance variables.

We use this example to investigate whether modelling tails in the final layer of a NF is superior to modelling the tails in the base distribution. Of particular interest is how performance varies with dimensionality $d$ and tail weight $\nu$. Appendix

F shows that neural network regression can perform poorly under heavy tailed inputs. Here we investigate whether a similar finding holds in the setting of NFs.

Some NF methods we test involve fixed tail weight parameters (either $\nu$ for a Student's T base distribution or $\lambda_+, \lambda_-$ for our transformation), and for this experiment we fix these to their known true values. This means our analysis is not confounded by the difficulty of estimating these parameters. The tail weights are known since the marginal density for each $X_i$ can be shown to be asymptotically proportional to $x_i^{-\nu-1}$. Note that only $\lambda_+, \lambda_-$ in (3) are fixed, not $\mu$ and $\sigma$.

**Flow Architectures** We investigate a selection of NF methods, including several which aim to address extremes. To conduct a fair comparison, we maintain as much consistency between the flow architectures as possible.

A baseline method, **normal**, uses a $d$-dimensional isotropic Gaussian base distribution. This is followed by an autoregressive rational quadratic spline (RQS) layer then an autoregressive affine layer. The latter should be capable of capturing linear dependency, while the former can adjust the shape of the body, but not the tails of the distribution.

Our proposed approach, tail transform flow, modifies the architecture just described by adding an additional layer for dealing with the tails, as described in Section 3. **TTF** trains the tail parameters alongside the other parameters. **TTFfix** is a 2-stage approach, which fixes the tail parameters to the known true values.

Marginal tail adaptive flows (**mTAF**) have the same architecture as **normal**, but use a Student's $T$ base distribution, as detailed in Section 1.1. The degrees of freedom are fixed to the correct tail parameters. Generalised tail adaptive flows (**gTAF**) differ in that the degrees of freedom are optimised alongside all other parameters during the training procedure.

As further variations on **normal**, we consider two alternative base distributions suggested by Amiri et al. (2022) – Gaussian mixture (**m_normal**) and Generalised Normal (**g_normal**). More details are provided in Appendix G.

We also consider COMET flows (**COMET**), as detailed in Section 1.1. For the normalizing flow part of this method we use the same architecture as **normal**. Our analysis is based on the code of McDonald et al. (2022) with some improvements to implementation details (needed to run more complicated examples later). COMET is a 2-stage approach, which involves estimating tail parameters in the first stage, so again we fix these to the known correct values.

We also tested a variant suggested by a reviewer, **TTF_tBase**. This combines TTF with a Student's $T$ base distribution with trainable degrees of freedom. It performed worse than both TTF and TTFfix, so is omitted from the main paper results

Table 1: Density estimation results on synthetic example for $d = 50$. Each entry is a mean value of negative test log likelihood per dimension across 10 repeated experiments, with the standard error in brackets. Bold indicates methods whose mean log likelihood differs from the best mean by less than 2 standard errors (of the best mean). A dash indicates potential unstable optimisation (at least one repeat had a final loss above 1e5).

| Flow | $\nu = 0.5$ | $\nu = 1$ | $\nu = 2$ |
|---|---|---|---|
| normal | - | - | 2.02 (0.01) |
| m_normal | - | - | 2.02 (0.00) |
| g_normal | - | - | 2.01 (0.00) |
| gTAF | 7.49 (0.38) | 2.65 (0.01) | 1.99 (0.00) |
| TTF | **3.68 (0.00)** | **2.54 (0.00)** | 1.98 (0.00) |
| mTAF | 5.22 (0.04) | 2.62 (0.01) | 1.98 (0.00) |
| TTFfix | **3.68 (0.00)** | **2.54 (0.00)** | 1.98 (0.00) |
| COMET | 3.74 (0.00) | 2.55 (0.00) | **1.97 (0.00)** |

for brevity. Its results appear in Appendix J.1.

**Experimental Details** We run 10 repeats for each flow/target combination. Each repeat samples a new set of data, with 5000 observations. which is split in proportion 40/20/40, to give training, validation and test sets respectively. We train using the Adam optimiser with a learning rate of 5e-3. We use an early stopping procedure, stopping once there has been no improvement in validation loss in 100 epochs, and returning the model from the epoch with best validation loss. Optimisation loss plots were also visually inspected to confirm convergence. The selected model was then evaluated on the test set to give a negative test log likelihood per dimension.

**Results** Table 1 shows a selection of results with $d = 50$. See Appendix J.1 for other $d$ and $\nu$ values. The methods not specifically designed to permit GPD tails (**normal**, **m_normal**, **g_normal**) are the worst performing, often not converging at all for $\nu \leq 1$. However, the difference between methods is small for $\nu = 2$, and all methods are similar for $\nu = 30$ (near-Gaussian tails—results in appendix).

For particularly heavy tails ($\nu \leq 1$), the best performing methods are TTF, TTFfix and COMET: the approaches which model tails in the final transformation. We did not detect a significant difference between the two TTF methods, but they both outperform COMET. A similar pattern is present for other choices of $d$ (results in appendix).

It is interesting that COMET performs competitively here, despite not inducing Fréchet tails. A possible reason is that COMET can produce log-normal tails (see Appendix C.2), and these have similar sub-asymptotic properties to GPD tails (Nair et al., 2022).

Another question is whether it's better to fix the tail parameters or optimise them. Table 1 shows that mTAF outperforms gTAF, so fixing tail parameters is better for heavy tailed base distribution methods. However there is no significant difference between TTF methods. A further investigation of the results suggested by a reviewer does suggest possible advantages of TTF which might become more significant in other settings. See Appendix J.2 for details.

## 4.2. Density Estimation with Real Data

This section investigates density estimation for several real datasets with extreme values, covering insurance, financial and weather applications. Three are taken from Liang et al. (2022); Laszkiewicz et al. (2022) and one is novel to this paper. Appendix I has more information about the datasets and standard preprocessing applied before density estimation.

**Flow Architectures**   In this experiment we compare NF architectures which performed reasonably well in Section 4.1: TTF, TTFfix, mTAF, gTAF and COMET.

For most examples we reuse the NF architectures described in Section 4.1 with a slight alteration. We add trainable linear layers based on the LU factorisation (Oliva et al., 2018). For TTF methods, this immediately precedes the TTF layer. Otherwise, this is the final layer. We found adding LU layers greatly improved empirical performance.

For the most complex dataset (CLIMDEX) we use a deeper architecture, as used in Laszkiewicz et al. (2022), to permit fair comparison with their results. This has 5 RQS layers, alternated with LU layers. We use this architecture for mTAF and gTAF, and modify it for TTF, TTFfix and COMET in the same way as described in Section 4.1.

**Experimental Details**   For all examples we run 10 repeats for each flow/target combination. For most examples we train for 400 epochs using the Adam optimiser with a learning rate of 5e-4. The exception is CLIMDEX, where we follow the more complex training setup of Laszkiewicz et al. (2022) (e.g. cosine annealing, more optimisation steps) to permit fair comparison with their results.

**Results**   The results are presented in Table 2. TTF outperforms previous NF methods for density estimation: it is the best performing method for 3 examples, and second best to TTFfix for the other. The performance of TTFfix is more variable: it is the best performing in one example, second best to TTF in two examples, but the worst performing in the remaining example.

Overall these results show TTF methods provide an improvement in density estimation performance, and that the additional freedom of TTF to learn tail behaviour compared to TTFfix (e.g. it can learn tail asymmetry) is beneficial in some cases, especially the most complicated dataset, CLIMDEX.

Also, estimating tail parameters for each marginal involves extra compute costs, especially for high dimensional datasets such as CLIMDEX, where the cost was comparable to fitting the normalizing flow (although easy to parallelise).

## 4.3. Variational Inference for Artificial Target

As a proof-of-concept investigation of variational inference, we perform experiments using (4) as a target distribution. Recall that in Section 4.1 we used this to generate synthetic data for density estimation. This model allows us to easily vary tail weights ($\nu$) and number of nuisance variables ($d$).

**Flow Architectures**   We compare four flows which exhibited good performance in Section 4.1: **TTF**, **TTFfix**, **mTAF** and **gTAF**. While COMET flows also performed well in the density estimation setting, they involve making kernel density estimates of marginal distributions, which has no obvious equivalent for the VI setting. So we do not include them in this comparison. As in Section 4.1, for TTFfix and mTAF we fix the tail parameters to the known correct values.

Although mTAF and gTAF were proposed as density estimation methods, it is straightforward to use these heavy tail base distribution methods for this VI task. As noted earlier, gTAF (Laszkiewicz et al., 2022) is equivalent to ATAF (Liang et al., 2022) in the context of variational inference, and trains the tail parameters alongside the other parameters. It is difficult to apply mTAF to VI in general, due to the difficulty of estimating the tail parameters, but in this case we have theoretical correct values.

**Experimental Details**   We take $d \in \{5, 10, 50\}$ and $\nu \in \{0.5, 1, 2, 30\}$ (heavier than Cauchy; Cauchy; lighter than Cauchy; close to Gaussian). We run 5 repeats for each flow/target combination. Each repeat uses 10,000 iterations of the Adam optimiser with learning rate 1e-3. Optimisation loss plots were visually inspected to confirm convergence.

**Diagnostics**   We measure the accuracy of our fitted variational density using two diagnostics based on importance sampling and described in Appendix H: $ESS_e$ and $\hat{k}$.

**Results**   Table 3 reports our results. TTFfix has best $ESS_e$ for heavier tails ($\nu \leq 2$). TTF produces slightly worse values, with mTAF and gTAF worse than either TTF method, especially for the $\nu \leq 1, d = 50$ cases ($ESS_e < 0.1$). For $\nu = 30$ the best results are for mTAF, but all methods do well ($ESS_e > 0.7$). All methods aside from gTAF achieve useful approximations ($\hat{k}$ below 0.7), in every setting where $\nu \geq 1$. In the most challenging setting—$\nu = 0.5, d = 50$—both TTF method achieve good $\hat{k}$ values.

Table 2: Real data density estimation results. Each entry is a mean test negative log likelihood over 10 trials, with standard deviation reported in brackets. Values marked with * are those reported by Laszkiewicz et al. (2022). Results within one standard deviation of the best mean are highlighted in bold.

| Model | Insurance | Fama 5 | S&P 500 | CLIMDEX |
|---|---|---|---|---|
| normal | 1.41 (0.03) | 4.65 (0.01) | 334.01 (1.02) | -2101.91 (9.44)* |
| gTAF | 1.41 (0.03) | 4.68 (0.01) | 321.81 (0.55) | -2113.48 (7.93)* |
| TTF | **1.37 (0.02)** | **4.61 (0.01)** | 317.56 (0.56) | **-2214.28 (13.25)** |
| mTAF | 1.52 (0.03) | 4.90 (0.03) | 322.98 (0.33) | -2121.38 (10.91)* |
| TTFfix | **1.38 (0.01)** | 4.63 (0.01) | **314.84 (0.46)** | -2090.91 (11.90) |
| COMET | 1.41 (0.03) | 4.63 (0.01) | 324.38 (0.58) | -2118.60 (8.61) |

Overall the results show that, as before, it is advantageous to model tails in the final transformation (TTF, TTFfix). Fixing tail parameters improves performance of TTFfix over TTF, perhaps because optimisation is more stable in this case.

## 5. Conclusion

Most current methods for modelling extremes with NFs use heavy tailed base distributions. We demonstrate this can perform poorly, due to extreme inputs to neural networks causing slow convergence. We present an alternative: use a Gaussian base distribution and a final transformation which can induce heavy tails. We propose the TTF transformation, equation (3), and prove it can indeed produce heavy tails. For density estimation, our TTF methods outperform existing methods on real and synthetic data. Jointly learning the tail parameters and the rest of the flow is usually the best approach. However for variational inference, fixing the tail parameters is beneficial, perhaps by improving optimisation stability. Our approaches do not damage performance in examples without heavy tails – here all types of NF we considered perform well.

### 5.1. Limitations and Future Work

An arguable limitation of our transformation is that it **always** converts Gaussian tails to heavy tails. Hence it cannot produce exactly Gaussian tails. It is not clear whether this is detrimental in practice, and it does not prevent good performance in all our experiments.

Another potential limitation is that the TTF transformation affects both the body and tail of the output distribution. The preceding layers of the flow must learn to model the body of the distribution, and also adapt to the final layer. It would be appealing to decouple the transformations, so that they can concentrate separately on the body and tail, which could make optimisation easier. This could be achieved by somehow ensuring the final transformation is approximately the identity in the body region.

Possible future work is to design methods which incorporate more extreme value theory properties. For instance, our method doesn't explicitly allow tail dependence (Coles et al., 1999). This would be an interesting future direction e.g. by adapting the manifold copula method of (McDonald et al., 2022). Also, it would be interesting to design multivariate transformations with the max-stable property (Coles, 2001).

We found that fixing the tail parameters to known values performed well in our VI example on an artificial target. However in real VI applications, the tail parameters aren't known, motivating methods to estimate them e.g. using static analysis of probabilistic programs (Liang et al., 2024). Also, in exploratory work on real VI applications, we found it difficult to improve VI results using our method. A possible reason is that our experiments only show major improvements for especially heavy tails ($\nu \leq 1$ in Table 3), which we found to be uncommon for real posteriors.

A potential future application is simulation based inference. Here **neural likelihood estimation** methods (Papamakarios et al., 2019) use NF density estimation to estimate a likelihood function from simulated data. Our approach could improve results when the simulated data is heavy tailed.

Finally, it would be interesting to use our tail transformation with other generative models. Here we briefly discuss differential equation based methods e.g. continuous normalizing flows, diffusion models, flow matching (Lipman et al., 2023). These sample $x(0)$ from a base distribution such as $\mathcal{N}(0, I)$, apply a differential equation $\frac{dx}{dt} = u(x, t)$ (or an SDE also including a diffusion term), and output $x(1)$. The vector field $u$ is a neural network, trained to produce a desired output distribution. Our work suggests it would be difficult to train a vector field for heavy tailed targets. To see this, suppose we desire output $x(1) = x^*$. By continuity, we need $x(1 - \delta) \approx x^*$ for small $\delta$. So we must evaluate $u$ for input similar to $x^*$, which can be extreme for a heavy tailed target. We've argued neural networks are hard to train with extreme inputs. This motivates using our tail transform as a final transformation to $x(1)$, effectively lightening the tails of the target, avoiding extreme $x^*$ values. Either joint training or a two-stage approach could be tried, similar to TTF and TTFfix.

Table 3: Variational inference results for artificial target. Each entry is a mean value across 5 repeated experiments. For $ESS_e$ columns, bold indicates the method with best value for each $d$. For $\hat{k}$ columns, bold indicates any value below 0.7.

| $d$ | Flow | $\nu = 0.5$ | | $\nu = 1$ | | $\nu = 2$ | | $\nu = 30$ | |
|---|---|---|---|---|---|---|---|---|---|
| | | $ESS_e$ | $\hat{k}$ | $ESS_e$ | $\hat{k}$ | $ESS_e$ | $\hat{k}$ | $ESS_e$ | $\hat{k}$ |
| 5 | gTAF | 0.47 | 0.89 | 0.53 | 0.93 | 0.90 | 0.79 | 0.97 | **0.53** |
| | TTF | 0.59 | **0.67** | 0.83 | **0.40** | 0.89 | **0.43** | 0.98 | **0.18** |
| | mTAF | 0.00 | 2.25 | 0.10 | **0.18** | 0.52 | **-0.18** | **0.99** | **0.31** |
| | TTFfix | **0.62** | 0.82 | **0.97** | **0.37** | **0.98** | **0.35** | 0.98 | **0.19** |
| 10 | gTAF | 0.19 | **0.64** | 0.28 | 1.05 | 0.74 | 0.89 | 0.96 | **0.28** |
| | TTF | 0.40 | 0.74 | 0.90 | **0.36** | 0.92 | **0.36** | 0.96 | **0.20** |
| | mTAF | 0.00 | 3.13 | 0.09 | **0.31** | 0.52 | **-0.37** | 0.98 | **0.25** |
| | TTFfix | **0.45** | 0.88 | **0.93** | **0.33** | **0.95** | **0.32** | 0.97 | **0.07** |
| 50 | gTAF | 0.02 | 0.90 | 0.08 | 0.82 | 0.39 | 0.83 | 0.84 | **0.13** |
| | TTF | 0.39 | **0.51** | 0.57 | **0.25** | 0.61 | **0.24** | 0.79 | **0.14** |
| | mTAF | 0.00 | 7.92 | 0.02 | **0.57** | 0.41 | **0.03** | 0.90 | **0.17** |
| | TTFfix | **0.43** | **0.53** | **0.75** | **0.18** | **0.81** | **0.18** | 0.85 | **0.08** |

## Impact Statement

This paper presents work whose goal is to advance the field of Machine Learning. There are many potential societal consequences of our work, none which we feel must be specifically highlighted here.

**Acknowledgements** Thanks to Miguel de Carvalho, Seth Flaxman, Iain Murray, Ioannis Papastathopoulos, Scott Sisson, Jenny Wadsworth, Peng Zhong and anonymous reviewers for helpful discussions. Tennessee Hickling is supported by a PhD studentship from the EPSRC Centre for Doctoral Training in Computational Statistics and Data Science (COMPASS). A preliminary version of this work appeared as Hickling & Prangle (2023).

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

## A. Gaussian mixtures are light tailed

As suggested by a reviewer, here we prove that Gaussian mixtures are light tailed. Then, by Theorem 1.2, using a Gaussian mixture base distribution with a Lipschitz NF produces light tailed output. Amiri et al. (2022) suggest this combination (reviewed in Section 1.1.3), and we consider it in our experiments under the name m_normal.

Suppose the real valued random variable $Z$ has density

$$q(z) = \sum_{i=1}^{m} w_i q_i(z),$$

where $\sum_{i=1}^{m} w_i = 1$, $w_i \in (0,1)$ and $q_i(z)$ is a $Z_i \sim \mathcal{N}(\mu_i, \sigma_i^2)$ density. The moment generating function of $Z_i$ is

$$M_{Z_i}(\lambda) = \mathbb{E}[\exp(\lambda Z_i)] = \exp[\lambda \mu_i + \lambda^2 \sigma_i^2 / 2].$$

Hence the moment generating function of $Z$ is

$$M_Z(\lambda) = \mathbb{E}[\exp(\lambda Z)] = \sum_{i=1}^{m} w_i M_{Z_i}(\lambda) = \sum_{i=1}^{m} w_i \exp[\lambda \mu_i + \lambda^2 \sigma_i^2 / 2].$$

So $M_Z(\lambda) < \infty$ for all $\lambda > 0$, and $Z$ is light tailed under Definition 1.1.

In this paper we concentrate on univariate properties. However, note that for $Z$ which is a mixture of multivariate Gaussians, we can consider any univariate random variable of the form $Z' = a^T Z$ (with $a^T a \neq 0$). Since $Z'$ is clearly a mixture of univariate Gaussians, it is light tailed by the argument above.

## B. TTF details

This appendix gives more details of our TTF transformation:

$$R(z; \lambda_+, \lambda_-) = \mu + \sigma \frac{s}{\lambda_s} [\text{erfc}(|z|/\sqrt{2})^{-\lambda_s} - 1].$$

Recall that $s = \text{sign}(z)$ and

$$\lambda_s = \begin{cases} \lambda_+ & \text{for } s = 1, \\ \lambda_- & \text{for } s = -1. \end{cases} \tag{5}$$

First, Section B.1 reviews some background on the complementary error function. Then Section B.2 presents a motivation for the transformation, and Section B.3 discusses some of its properties.

### B.1. Complementary error function

For $z \in \mathbb{R}$, the error function and complementary error function are defined as

$$\text{erf}(z) = \frac{2}{\sqrt{\pi}} \int_0^z \exp(-t^2) dt,$$

$$\text{erfc}(z) = 1 - \text{erf}(z).$$

For large $z$, $\text{erfc}(z) \approx 0$, so it can be represented accurately in floating point arithmetic.

Note that

$$F_N(z) = \frac{1}{2}(1 + \text{erf}[z/\sqrt{2}]), \tag{6}$$

where $F_N(z)$ is the $\mathcal{N}(0,1)$ cumulative distribution function. This implies

$$\text{erfc}^{-1}(x) = -F_N^{-1}(x/2)/\sqrt{2}. \tag{7}$$

Efficient numerical evaluation of $\text{erfc}$ and its gradient is possible, as it is a standard special function (Temme, 2010), implemented in many computer packages. For instance PyTorch provides the `torch.special.erfc` function. Later we also require $\text{erfc}^{-1}$, which is less commonly implemented directly. However using (7) we can compute $\text{erfc}^{-1}$ using the more common standard normal quantile function $F_N^{-1}$. For instance PyTorch implements this as `torch.special.ndtri` (although this can have problems for small inputs—see Appendix G.5).

## B.2. Motivation

This section motivates the TTF transformation by sketching how it can be derived from some simpler transformations. This gives an informal derivation that $R$ transforms $\mathcal{N}(0, 1)$ tails to GPD tails. The formal version, Theorem C.9, is presented later and also appropriately generalises the result to $\mathcal{N}(\mu, \sigma^2)$ inputs.

**GPD transform**    Section 2.4 motivates the transformation $P : [0, 1] \to \mathbb{R}^+$ given by the GPD quantile function

$$P(u; \lambda) = \frac{1}{\lambda}[(1 - u)^{-\lambda} - 1], \tag{8}$$

where $\lambda > 0$ is the shape parameter. Using $P$ transforms a distribution with support $[0, 1]$ to one on $\mathbb{R}^+$ with tunable tail weight.

We note that (8) is also valid for $\lambda < 0$ and maps $[0, 1]$ to $[0, \frac{1}{\lambda}]$. Here the image of the transformation depends on the parameter values, which is not a useful property for a normalizing flow. Hence we do not consider negative $\lambda$ in our work.

**Two tailed transform**    We can extend (8) to a transformation $Q : [-1, 1] \to \mathbb{R}$,

$$Q(u; \lambda_+, \lambda_-) = \frac{s}{\lambda_s}[(1 - |u|)^{-\lambda_s} - 1]. \tag{9}$$

Here $\lambda_+ > 0, \lambda_- > 0$ are shape parameters for the positive and negative tails. We use the notation $s = \text{sign}(u)$, with $\lambda_s$ defined as in (5).

Using $Q$ transforms a distribution with support $[-1, 1]$ to one on $\mathbb{R}$ with tunable weights for both tails.

**Real domain transform**    We would like a transformation $R : \mathbb{R} \to \mathbb{R}$ which can transform Gaussian tails to GPD tails. Consider $z \in \mathbb{R}$, and let $u = 2F_N(z) - 1$, where $F_N$ is the $\mathcal{N}(0, 1)$ cumulative distribution function. Then $u \in [-1, 1]$ and we can output $Q(u)$. A drawback is that large $|z|$ can give $u$ values which are rounded to $\pm 1$ numerically.

Using (6) shows that

$$1 - |u| = \text{erfc}(|z|/\sqrt{2}), \tag{10}$$

so there is a standard special function to compute $1 - |u|$ directly. Large $|z|$ gives $1 - |u| \approx 0$, which can be represented to high accuracy in floating point arithmetic, avoiding catastrophic rounding which could result from working directly with $u$.

Substituting (10) into (9) and adding location and scale parameters results in our TTF transformation $R$.

## B.3. Properties

Here we derive several properties of the TTF transformation, $R$. These include expressions which are useful in an automatic differentiation implementation of $R$ for NFs: $R^{-1}$ and derivatives of $R$ and $R^{-1}$.

**Forward transformation**    The derivative of $R$ with respect to $z$ is given by

$$\frac{\partial R}{\partial z}(z; \mu, \sigma, \lambda_+, \lambda_-) = \sigma\sqrt{\frac{2}{\pi}}\exp(-z^2/2)\,\text{erfc}(|z|/\sqrt{2})^{-\lambda_s-1}. \tag{11}$$

All of these factors are positive, which implies that for all parameter settings $\frac{\partial R}{\partial z}(z) > 0$, so the transformation is monotonically increasing.

The transformation is continuously differentiable at $z = 0$ as

$$\frac{\partial R}{\partial z}(0; \mu, \sigma, \lambda_+, \lambda_-) = \sigma\sqrt{\frac{2}{\pi}}. \tag{12}$$

This has no dependence on the $\lambda_+, \lambda_-$ and is the limit as $z \to 0$ from above or below.

**Inverse transformation** Define $y = \lambda_s |(x - \mu)/\sigma| + 1$. Then the inverse transformation is

$$R^{-1}(x; \mu, \sigma, \lambda_+, \lambda_-) = s\sqrt{2}\,\mathrm{erfc}^{-1}(y^{-1/\lambda_s}). \tag{13}$$

Note here we can take $s = \mathrm{sign}(x - \mu)$. The gradient is

$$\frac{\partial R^{-1}}{\partial x}(x; \mu, \sigma, \lambda_+, \lambda_-) = \frac{1}{\sigma}\sqrt{\frac{\pi}{2}} y^{-1/\lambda_s - 1} \exp\left(\mathrm{erfc}^{-1}(y^{-1/\lambda_s})^2\right). \tag{14}$$

# C. Asymptotic results

This appendix proves asymptotic results on the tails produced by COMET flows and our TTF transformation. Our main aim to to prove whether or not the distributions are in the Fréchet domain of attraction $\Theta_\lambda$ (see Definition 2.1). We will also find the tail behaviour in detail for some special cases.

Throughout this appendix we use the following asymptotic notation. In particular, $\sim$ is **not** used to describe probability distributions, as it is elsewhere in the paper.

**Definition C.1.** For functions $f : \mathbb{R} \to \mathbb{R}^+$ and $g : \mathbb{R} \to \mathbb{R}^+$, we write

1. $f(x) = O(g(x))$ if $\limsup_{x \to \infty} f(x)/g(x) < \infty$,

2. $f(x) \sim g(x)$ if $\lim_{x \to \infty} f(x)/g(x) = 1$.

## C.1. Background

The material in this section will be used in the proofs later in this appendix. The first results are on regularly varying functions, based on the presentation in Nair et al. (2022).

**Definition C.2.** A function $f : \mathbb{R}^+ \to \mathbb{R}^+$ is **regularly varying** of index $\rho \in \mathbb{R}$ if

$$\lim_{x \to \infty} f(kx)/f(x) = k^\rho$$

for all $k > 0$. When $\rho = 0$, $f(x)$ is **slowly varying**.

**Theorem C.3** (Nair et al., Lemma 2.7)**.** *For slowly varying $f(x)$,*

$$\lim_{x \to \infty} x^\rho f(x) = \begin{cases} 0 & \text{if } \rho < 0, \\ \infty & \text{if } \rho > 0. \end{cases}$$

**Theorem C.4** (Nair et al., Theorem 2.8)**.** *A function $f : \mathbb{R}^+ \to \mathbb{R}^+$ is regularly varying of index $\rho$ if and only if*

$$f(x) = x^\rho \ell(x)$$

*for some slowly varying $\ell$.*

The following result is an immediate corollary of Theorem 7.5(i) of Nair et al.

**Theorem C.5** (Nair et al., corrolary)**.** *Consider a real-valued random variable $X$ with density $q(x)$. Its distribution is in $\Theta_\lambda$ if and only $q$ is regularly varying of index $-1 - \lambda$.*

Finally, we present a result of Liang et al. (2022).

**Definition C.6.** Let $\overline{\mathcal{E}^2}$ be the set of random variables $X$ such that for some $\alpha > 0$

$$\Pr(|X| \geq x) = O(\exp[-\alpha x^2]).$$

Note that $\overline{\mathcal{E}^2}$ includes Gaussian distributions, and does not intersect with $\Theta_\lambda$ for any $\lambda > 0$.

The following result is a special case of Theorem 3.2 of Liang et al., and is a more precise extension of Theorem 1.2.

**Theorem C.7** (Liang et al. special case)**.** $\overline{\mathcal{E}^2}$ *is closed under Lipschitz transformations.*

Of interest is a Gaussian base distribution transformed by a Lipschitz transformation, such as most standard NFs. The result shows that the output is in $\overline{\mathcal{E}^2}$, and not the Fréchet domain of attraction.

### C.2. COMET flow tails

Recall that COMET flows (McDonald et al., 2022), reviewed in Section 1.1, use a Gaussian base distribution followed by a generic normalizing flow. Then each component of the output is transformed by $h \circ g$, where $g$ is the logistic transformation and $h$ is a GPD quantile function. The following result describes the resulting tails.

**Theorem C.8.** *Let $Z$ be a real random variable with density $p(z)$. Define $X = h \circ g(Z)$ where*

$$g(z) = \frac{1}{1 + \exp(-z)}, \quad h(z) = \frac{1}{\lambda}\left[(1 - z)^{-\lambda} - 1\right].$$

*Let $p_N(z; \mu, \sigma^2)$ denote a $\mathcal{N}(\mu, \sigma^2)$ density. Then:*

1. *$Z \in \overline{\mathcal{E}^2}$ (see Definition C.6) implies $X \notin \Theta_\lambda$ (the Fréchet domain of attraction).*

2. *For $p(z) \sim p_N(z; 0, \sigma^2)$, $X$ has density $q(x) \sim q_{LN}(x; 0, \lambda^2\sigma^2)$, a log-normal density with location zero and scale $\lambda^2\sigma^2$.*

Statement 1 implies that the tails produced by COMET flows are not suitably heavy tailed under most standard NFs. The argument is as follows. Recall that (Jaini et al., 2020; Liang et al., 2022) give details showing that most standard NFs are Lipschitz. Theorem C.7 shows that applying a Lipschitz transformation to a Gaussian base distribution produces a distribution in $\overline{\mathcal{E}^2}$. Then statement 1 shows that COMET flows do not map this to the Fréchet domain of attraction, which is the desired property by Definition 2.1.

Statement 2 shows that in one particular case, the resulting tails are log-normal. While the log-normal distribution is heavy tailed under Definition 1.1, it is not in the Fréchet domain of attraction (see e.g. Embrechts et al., 2013, Example 3.3.31). Hence it cannot capture the full range of asymptotic tail behaviours. For instance, all moments exist for a log normal distribution but not for a GPD.

*Proof.* Let $x = h(v), v = g(z)$. It's straightforward to derive that

$$z \sim \lambda^{-1} \log x, \tag{15}$$
$$x \sim \exp(\lambda z)/\lambda, \tag{16}$$
$$g'(z) \sim (\lambda x)^{-1/\lambda}, \tag{17}$$
$$h'(v) \sim (\lambda x)^{1+1/\lambda}. \tag{18}$$

The change of variables theorem gives

$$q(x) = \frac{p(z)}{|g'(z)||h'(v)|} = \frac{p(z)}{\lambda x}. \tag{19}$$

**Statement 1** We'll prove the contrapositive: $X \in \Theta_\lambda$ implies $Z \notin \overline{\mathcal{E}^2}$.

By Theorems C.4 and C.5, $X \in \Theta_\lambda$ implies $q(x) = x^{-\lambda-1}\ell(x)$ with $\lambda > 0$ and slowly varying $\ell$. Thus

$$\begin{aligned} p(z) &\sim q(x)\lambda x && \text{from (19)} \\ &= x^{-\lambda-1}\ell(x)\lambda x && \text{expression for } q(x) \\ &\sim \exp(-\lambda[\lambda + 1]z)\lambda^{\lambda+2}x\ell(x). && \text{from (16)} \end{aligned}$$

Let $s(z) = \lambda(\lambda + 1)\exp(-\lambda[\lambda + 1]z)$. Then

$$\frac{p(z)}{s(z)} \sim Dx\ell(x),$$

with $D = \lambda^{\lambda+1}/(\lambda + 1)$. By Theorem C.3 this ratio converges to $\infty$.

It follows that there exists $z_0$ such that $z > z_0$ implies $p(z) > s(z)$. Thus for $z > z_0$:

$$\Pr(Z > z) = \int_z^\infty p(t)dt > \int_z^\infty s(t)dt = \exp(-\lambda[\lambda + 1]z).$$

This contradicts $\Pr(|Z| > z) = O(\exp[-\alpha z^2])$ for some $\alpha > 0$, so $Z \notin \overline{\mathcal{E}^2}$.

**Statement 2** From (19), $q(x) \sim p_N(z; \mu, \sigma^2)/\lambda x$. Substituting in the Gaussian density and (15) gives

$$q(x) \sim r(x) = Ax^{-1+B} \exp[-C(\log x)^2],$$

where $A = \frac{1}{\lambda \sigma (2\pi)^{1/2}} \exp[-\frac{\mu^2}{2\sigma^2}]$, $B = \frac{\mu}{\lambda \sigma^2}$ and $C = \frac{1}{2\lambda^2 \sigma^2} > 0$. For $\mu = 0$, $r(x)$ is a log-normal $LN(0, \lambda^2 \sigma^2)$ density, as required.

$\square$

**Remark** The proof of statement 1 suggests that $Z$ with an exponential distribution would produce output in the Fréchet domain of attraction. This motivates using a Laplace distribution as a base distribution for a COMET flow. A drawback is the lack of smoothness of the Laplace density at the origin.

### C.3. TTF tails

Our transformation $R$ is defined in (3). Here we consider a simplified version $S : \mathbb{R}^+ \to \mathbb{R}^+$,

$$S(z; \lambda) = \frac{1}{\lambda}[\text{erfc}(z/\sqrt{2})^{-\lambda} - 1]. \tag{20}$$

This modifies $R$ by setting the location parameter to zero and scale parameter to one, and only considering the upper tail. To simplify notation it uses $\lambda$ in place of $\lambda_+$. We provide the following result on tails produced by this transformation.

**Theorem C.9.** *Let $Z$ be a random variable with density $p(z) \sim p_N(z; \mu, \sigma^2)$, a $\mathcal{N}(\mu, \sigma^2)$ density. Define $X = S(Z; \lambda)$. Then:*

1. *The distribution of $X$ is in the Fréchet domain of attraction with shape parameter $\lambda \sigma^2$.*

2. *For $\mu = 0, \sigma = 1$, $X$ has density $q(x) \sim q_{GPD}(x; \lambda, 2^{1/(2+1/\lambda)})$, a GPD density with shape parameter $\lambda$ and scale $2^{1/(2+1/\lambda)}$ (see Definition C.10 below).*

The result for the lower tail is a simple corollary. Also, the result is unaffected by adding a final location and scale transformation—as in our full TTF transformation $R$—except that the latter will modify the scale parameter in statement 2.

As argued in Section 3.1, most NFs use a standard normal base distribution, and are capable of producing the identity transformation. So statement 2 of Theorem C.9 shows that composing them with $R$ can produce output which has heavy tails with parameterised weights. Statement 1 provides robustness: the result remains true when the input to $R$ is a Gaussian with arbitrary location and scale parameters.

### C.3.1. LEMMAS

Here we present two lemmas which are used in the proof of Theorem C.9.

**Definition C.10.** A GPD with shape $\lambda > 0$, location 0, and scale $\sigma > 0$ has density

$$q_{GPD}(x; \lambda, \sigma) = \frac{1}{\sigma}(1 + \lambda x/\sigma)^{-1-1/\lambda}.$$

**Lemma C.11.** *For $k > 0$,*

$$kq_{GPD}(x; \lambda, 1) \sim q_{GPD}(x; \lambda, k^{-1/(2+1/\lambda)}).$$

The left hand side is a GPD density with scale 1, multiplied by a constant $k$. The lemma shows this is asymptotically equivalent to a GPD density with unchanged shape but modified scale.

*Proof.* Observe that

$$q_{GPD}(x; \lambda, \sigma) \sim \sigma^{-2-1/\lambda}(\lambda x)^{-1-1/\lambda}$$
$$\Rightarrow kq_{GPD}(x; \lambda, 1) \sim k(\lambda x)^{-1-1/\lambda} \sim q_{GPD}(x; \lambda, k^{-1/(2+1/\lambda)}).$$

$\square$

**Lemma C.12.** *Suppose $x = S(z; \lambda)$. Then for large $x$,*

$$z = [\tfrac{2}{\lambda} \log(\lambda x + 1)]^{1/2} + o(1), \tag{21}$$

$$z^2 = \tfrac{2}{\lambda} \log(\lambda x + 1) - \log \log(\lambda x + 1) + \log \tfrac{\lambda}{\pi} + o(1). \tag{22}$$

*Proof.* Let $F_N(\cdot)$ be the $\mathcal{N}(0, 1)$ cdf. The result is a corollary of the following asymptotic expansion from (Fung & Seneta, 2018), for $p \to 0$

$$F_N^{-1}(p) = -[-2 \log p - \log \log p^{-2} - \log 2\pi]^{1/2} \left( 1 + O \left[ \frac{\log |\log p|}{(\log p)^2} \right] \right).$$

By definition, $x = \tfrac{1}{\lambda}[\operatorname{erfc}(z/\sqrt{2})^{-\lambda} - 1]$. So $z = \sqrt{2} \operatorname{erfc}^{-1}(p)$ where $p = (1 + \lambda x)^{-1/\lambda}$. Using (7) gives $z = -F_N^{-1}(p/2)$. Hence we get

$$z = [\tfrac{2}{\lambda} \log(\lambda x + 1)]^{1/2} \left[ 1 + \frac{-\log \log(\lambda x + 1) + \log \tfrac{\lambda}{\pi} + o(1)}{\tfrac{2}{\lambda} \log(\lambda x + 1)} \right]^{1/2} \left( 1 + O \left[ \frac{\log \log(\lambda x + 1)}{[\log(\lambda x + 1)]^2} \right] \right).$$

Expanding the middle factor using a Taylor series and checking the order of the remaining terms gives (21). Squaring the asymptotic expansion for $z$ and checking the order of terms gives (22). $\qquad\square$

**Remark** A consequence of (22) which we will use later is:

$$z^2 = \eta(x) - \log \eta(x) + o(1), \tag{23}$$

$$\text{where} \quad \eta(x) = \tfrac{2}{\lambda} \log(\lambda x + 1) + \log \tfrac{2}{\pi}.$$

### C.3.2. PROOF OF THEOREM C.9

Let $x = S(z; \lambda)$. The change of variables theorem gives:

$$q(x) = p(z)/|S'(z; \lambda)|.$$

Recall that

$$p(z) \sim p_N(z; \mu, \sigma^2) = \frac{1}{(2\pi)^{1/2}\sigma} \exp\left[ -\tfrac{1}{2\sigma^2}(z - \mu)^2 \right],$$

and from (11),

$$S'(z; \lambda) = \sqrt{\frac{2}{\pi}} \exp(-z^2/2) \operatorname{erfc}(z/\sqrt{2})^{-\lambda-1}.$$

Hence

$$q(x) \sim \frac{1}{2} r(z; \mu, \sigma) \operatorname{erfc}(z/\sqrt{2})^{\lambda+1},$$

where

$$r(z; \mu, \sigma) = \frac{1}{\sigma} \exp\left[ -\frac{1}{2} \left\{ \frac{1}{\sigma^2}(z - \mu)^2 - z^2 \right\} \right].$$

Rearranging (20) gives $(1 + \lambda x)^{-1/\lambda} = \operatorname{erfc}(z/\sqrt{2})$. So

$$q(x) \sim \tfrac{1}{2} r(z; \mu, \sigma)(1 + \lambda x)^{-(1+1/\lambda)}. \tag{24}$$

Note that $r(z; 0, 1) = 1$. Then the proof of statement 2 concludes by applying Lemma C.11 to (24).

The remainder of the proof is for statement 1, and uses capital letters to represent constants with respect to $x$. We have

$$\log r(z; \mu, \sigma) = \tfrac{1}{2}(1 - \sigma^{-2})z^2 + Az + B.$$

Using Lemma C.12 gives

$$\log r(z; \mu, \sigma) = \tfrac{1}{\lambda}(1 - \sigma^{-2}) \log(\lambda x + 1) + C[\log(\lambda x + 1)]^{1/2} + D \log \log(\lambda x + 1) + E + o(1).$$

So

$$r(z; \mu, \sigma) = F(\lambda x + 1)^{(1-\sigma^{-2})/\lambda} \exp\left\{C[\log(\lambda x + 1)]^{1/2}\right\} [\log(\lambda x + 1)]^D (1 + o(1)).$$

Substituting into (24) gives

$$q(x) \sim \tfrac{1}{2}(1 + \lambda x)^{-1-\lambda^{-1}\sigma^{-2}} F \exp\left\{C[\log(\lambda x + 1)]^{1/2}\right\} [\log(\lambda x + 1)]^D (1 + o(1))$$

$$\Rightarrow q(x) = x^{-1-\lambda^{-1}\sigma^{-2}} \ell(x),$$

where

$$\ell(x) = G \exp\left\{C[\log(\lambda x + 1)]^{1/2}\right\} [\log(\lambda x + 1)]^D (1 + o(1))$$

can easily be checked to be a slowly varying function. Applying Theorems C.4 and C.5 shows that $Y$ is in the Fréchet domain of attraction with shape $\lambda\sigma^2$ as required.

## D. TTF Transform: Variations and Related Work

This appendix discusses variations on our TTF transform $R$, (3), and related work in the literature. For simplicity we compare to $R$ with location zero and scale one.

As well as those discussed below, we expect many other variations are possible with equivalent asymptotic properties, providing much scope for potential future study.

**Student's $T$ CDF Transform**    A straightforward alternative transformation which converts Gaussian tails to heavy tails is

$$R_{\text{cdf}} = F_T^{-1} \circ F_N$$

where $F_T$ and $F_N$ are cdfs of Student's $T$ and Gaussian distributions (both with location zero and scale one). Rather than use the Student's $T$ distribution for the base distribution, as in the prior work reviewed in Section 1.1.2, this moves the use of the $T$ distribution to the final transformation. Unfortunately, $F_T^{-1}$ does not have a closed form allowing automatic differentiation, so it lacks the flexibilty of TTF. In particular we cannot learn the tail weight parameters. However, $R_{\text{cdf}}$ could be applied in a two step procedure for density estimation similar to TTFfix (see Section 3.4). Exploratory work suggests this has comparable performance to TTFfix.

**Unit Scale Transformation**    Another alternative transformation is

$$R_{\text{alt}}(z; \lambda_+, \lambda_-) = \frac{s}{\lambda_s} \left[ \left\{ \frac{1}{2} \operatorname{erfc}(|z|/\sqrt{2}) \right\}^{-\lambda_s} - 1 \right].$$

This only differs from our TTF transform in that a factor of $1/2$ has been included. Equation 3.15 of Shaw et al. (2014) shows that for $z \to \infty$, $R_{\text{alt}}$ is asymptotically equivalent to $R_{\text{cdf}}$. Hence $R_{\text{alt}}$ converts $\mathcal{N}(0, 1)$ tails to GPD tails. In this case it has the advantage of producing output with scale one, which the TTF transformation does not do (see Theorem C.9, statement 2). However, the reason we don't use $R_{\text{alt}}$ is due to a disadvantage—there can be a discontinuity at $z = 0$, since

$$\lim_{z \to 0_+} R_{\text{alt}}(z; \lambda_+, \lambda_-) = (2^{\lambda_+} - 1)/\lambda_+,$$

$$\lim_{z \to 0_-} R_{\text{alt}}(z; \lambda_+, \lambda_-) = (2^{\lambda_-} - 1)/\lambda_-.$$

Another related result is from (Troshin, 2022). This paper investigates necessary and sufficient conditions for transformations to map $\mathcal{N}(0, 1)$ inputs to the Fréchet domain of attraction. This produces a family of functions which includes $R$ and $R_{\text{alt}}$—see Troshin's Remark 2. However, unlike our Appendix C, this work does not provide results for $\mathcal{N}(\mu, \sigma^2)$ inputs.

## E. Universality

Here we prove some universality properties relating to the TTF transformation, as referred to in Section 3.5, which also includes a broader discussion on universality.

## E.1. Background

We define universality for our purposes in Sections E.1–E.2 as follows.

**Definition E.1.** Let $\mathcal{S}$ be a set of transformations $s : \mathbb{R}^d \to \mathbb{R}^d$. Let $Z \sim \mathcal{N}(0, I)$ be the usual base distribution. We call $\mathcal{S}$ **universal** if for every random variable $X$ on $\mathbb{R}^d$ there is a sequence $s_n \in \mathcal{S}$ such that $s_n(Z) \to X$ in the sense of weak convergence as $n \to \infty$.

Universality has been proved for some NFs including SoS polynomial flows and Neural Autoregressive Flows (Huang et al., 2018; Jaini et al., 2019; Kobyzev et al., 2020).

For all these results the set of transformations $\mathcal{S}$ comprises NFs of a particular type with unbounded capacity. The sequence $s_n$ typically requires increasing capacity as $n \to \infty$. So Definition E.1 means that an **idealised** sequence of NFs can approximate any target distribution. In practice, capacity is limited and only an imperfect approximation is usually achievable.

For many other common NFs it is unknown whether universality holds: see Jaini et al., 2019 Table 1, but note progress has been made on coupling flows, discussed in Section E.3.

## E.2. TTF preserves universality

Let $\mathcal{S}$ be the set of transformations corresponding to a universal NF. Let $\mathcal{R}$ be the set of TTF transformations under all valid parameter choices. Let $\tilde{\mathcal{S}} = \{r \circ s : r \in \mathcal{R}, s \in \mathcal{S}\}$. We now prove that universality of $\mathcal{S}$ implies universality of $\tilde{\mathcal{S}}$. Note that $\mathcal{R}$ does not contain the identity, so the result is not immediate.

Fix any $\tilde{r} \in \mathcal{R}$ and some target $X$. Since $\mathcal{S}$ is universal, it contains a sequence $\{s_n\}$ such that $s_n(Z) \to \tilde{r}^{-1}(X)$. By the continuous mapping theorem (see e.g. Van der Vaart, 2000, Theorem 2.3), the continuity of $\tilde{r}$ implies $\tilde{r} \circ s_n(Z) \to X$, as required.

## E.3. Coupling flows

Draxler et al. (2024) investigate universality of coupling flows. They replace Definition E.1 by an alternative convergence metric specialised to the problem. Given a target random variable $X$ and a normalising flow $s$ acting on base random variable $Z \sim \mathcal{N}(0, I)$, it considers the closeness of $s^{-1}(X)$ to $\mathcal{N}(0, I)$ (in terms of the improvement possible by using $s \circ t$ where $t$ is a single coupling flow layer). See Definition 5.1 of Draxler et al. (2024) for full details.

Their main result – Theorem 5.4 of Draxler et al. 2024 – applies to any $X$ with full support, a continuous density and finite first and second moments. They show there exists a sequence of coupling flows $\{s_n\}$ achieving a convergence metric tending to zero.

The assumption that $X$ has finite first and second moments excludes sufficiently heavy tailed distributions from this result. This improves on earlier work such as Teshima et al. (2020) which looks at weak convergence in a bounded subset of $\mathbb{R}^d$, and so does not consider tails at all.

It is plausible that our TTF transformation can preserve the result of Draxler et al. (2024) and extend it to heavier tailed targets. Given some target $X$, the idea is to select $\tilde{r} \in \mathcal{R}$ such that $X' = \tilde{r}^{-1}(X)$ has finite first and second moments. That is, $\tilde{r}^{-1}$ acts to lighten tails of $X$ which are too heavy. Take the sequence $\{s_n\}$ from applying the result of Draxler et al. (2024) to $X'$. Then for target $X$ the sequence $\{\tilde{r} \circ s_n\}$ produces a convergence metric tending to zero.

However further results are needed to prove when a suitable $\tilde{r}$ exists. We leave this for future work.

# F. Neural Network Regression with Extreme Inputs

In Section 1, we claimed that neural network optimisation can perform poorly under heavy tailed input. This forms part of the motivation for proposing our methods. In Section 4.1 we illustrated this claim empirically for an example involving NFs. Here we provide further support in a simpler supervised learning example.

Our experiment considers the following simple regression problem

$$\{X_i\}_{i=1}^d \sim t_\nu, \quad Y \sim \mathcal{N}(X_d, 1).$$

Table 4: Neural network regression example results. Values are the median test MSE over 5 trials, with the maximum provided in brackets. (We use median as it's more robust than mean to extreme outliers. Maximum illustrates the presence and magnitude of such outliers.)

| $d$ | $\nu$ | Sigmoid activation | ReLU activation |
|---|---|---|---|
| | 30.0 | 1.01 (1.02) | 1.01 (1.04) |
| 5 | 2.0 | 1.74 (6.19) | 1.06 (1.08) |
| | 1.0 | 1.57e+03 (8.57e+06) | 3.45 (1.93e+03) |
| | 30.0 | 1.02 (1.02) | 1.02 (1.05) |
| 10 | 2.0 | 2.04 (7.28) | 1.03 (1.05) |
| | 1.0 | 4.13e+03 (1.09e+06) | 8.41 (49.6) |
| | 30.0 | 1.01 (1.04) | 1.08 (1.09) |
| 50 | 2.0 | 3.09 (5.26) | 1.27 (12.3) |
| | 1.0 | 1.03e+04 (1.73e+06) | 55.8 (6.59e+03) |
| | 30.0 | 1.02 (1.07) | 1.15 (1.2) |
| 100 | 2.0 | 3.41 (4.33) | 1.60 (1.74) |
| | 1.0 | 1.02e+05 (1.94e+05) | 336 (1.2e+03) |

The $d$-dimensional input is heavy tailed. The output equals one of the inputs plus Gaussian noise. The other inputs act as nuisance variables.

**Experimental Details**   We consider a number of tail weight ($\nu$) and dimension ($d$) combinations and generate 5000 train, validation and test samples for each. Our models are simple 2 layer multi-layer perceptrons with 50 nodes in each hidden layer. The models are optimised with Adam to minimise mean square error, selecting the model with smallest validation loss found during training. We consider both sigmoid and ReLU activations. This is to investigate whether the main issue is saturation of sigmoid activation functions. The experiment was repeated 5 times, each trial sampling a new set of data.

**Results**   Table 4 shows the results of the experiment. Both models perform well for $\nu = 30$ (light tailed inputs), worse for $\nu = 2$, and very badly for $\nu = 1$. Performance also decays as $d$ increases, but the effect is weaker. The ReLU activation function performs a little better overall, but can still result in very large MSE values.

Overall, the results demonstrate the failure of neural network methods to capture a simple relationship in the presence of heavy tailed inputs. The problem persists under a ReLU activation function, showing that it is not caused solely by saturation of the sigmoid activation function.

# G. Implementation Details

This appendix contains more details of how we implement our experiments in Section 4 of the main paper.

### G.1. Normalizing Flows

We use the `nflows` package (Durkan et al., 2020) to implement the NF models. This depends on PyTorch (Paszke et al., 2019) for automatic differentiation.

With the exception of the CLIMDEX example, our spline and autoregressive affine NF layers output the required transformation parameters from masked neural networks. These use 2 hidden layers, each with a width equal to the input dimension plus 10. Spline layers are configured to use a bounding box of $[-2.5, 2.5]$ with 5 bins.

For the more complicated CLIMDEX dataset, which requires more capacity, we reuse the architecture and tuning choices used in Laszkiewicz et al. (2022), as discussed in Section 4.2.

### G.2. Alternative Base Distributions

Standard NFs use a Gaussian base distribution, and a Student's $T$ base distribution is also popular for heavy tailed targets. In addition to these, we also tried two alternative base distributions suggested by Amiri et al. (2022) which we describe here.

The first is a Gaussian mixture model base distribution (**m_normal**) with 5 components (preliminary work found 10 or 15 components had no significant difference in performance). For each component we optimise the $d$ dimensional mean and diagonal covariance terms along with other parameters.

The second alternative base distribution is $d$ independent generalised normal distributions (**g_normal**). The generalised normal family does not have Pareto tails, so we cannot fix them to have the true tail weight. Instead, we optimise the shape parameter of each marginal.

### G.3. Tail Parameter Initialisation

For density estimation of heavy tailed data, the tail parameters must be initialised sufficiently high that large observations aren't mapped to vanishingly low probability regions of the base distribution. If this does occur, it is possible to get numerical overflow during optimisation. For variational inference a related problem can occur: if some tail parameters are too high, then we can sample points of very low target density and get numerical overflow. In practice, we find that initialising $\lambda$ uniformly from $[0.05, 1]$ provides sufficient stability.

For two-stage methods, Section 3.4 describes our initial stage of tail parameter estimation. Equivalent tail parameters are used for mTAF, TTFfix and COMET. In some cases, this estimation procedure finds that a marginal distribution is light tailed. In these cases, as mentioned in Section 3.3, we set the tail parameter to a small value, corresponding to a degree of freedom of $\nu = 1000$.

### G.4. Sampling Student's T

Consider sampling from a Student's $T$ distribution with $\nu$ degrees of freedom. Algorithm 1 implements this, while allowing the reparameterization trick i.e. differentiation with respect to $\nu$.

---
**Algorithm 1** Sampling Student's $T$
---
**Require:** Input: degrees of freedom $\nu$, threshold $\epsilon \geq 0$.
 1: Sample $g \sim \text{Gamma}(\nu/2, 1)$.
 2: Let $g' = \max(g, \epsilon)$.
 3: Sample $z \sim \mathcal{N}(0, 1)$.
 4: Return $z\sqrt{\frac{\nu}{2g'}}$.

---

Abiri & Ohlsson (2019) suggest this algorithm with $\epsilon = 0$, and comment on how to differentiate through a Gamma distribution. This method is implemented in PyTorch. However we found that taking the reciprocal of $g \approx 0$ can result in an overflow error. Hence we make a slight adjustment for numerical stability: we clamp $g$ to $\epsilon = 1\text{e-}24$.

### G.5. Inverse Transformation for Small Inputs

Equation (13) gives an expression for $R^{-1}(x)$, the inverse of our TTF transformation $R$, in terms of $\text{erfc}^{-1}(y^{-1/\lambda_s})$ where $y(x) = \lambda_s|(x - \mu)/\sigma| + 1$. However we experience numerical issues when implementing (13) for small $x$. Therefore for $y^{-1/\lambda_s} < 10^{-6}$ our code uses the approximation

$$\widehat{R^{-1}}(x) = s\left[\eta(x) - \log \eta(x)\right]^{\frac{1}{2}},$$

$$\text{where} \quad \eta(x) = \frac{2}{\lambda_s}\log y(x) + \log\frac{2}{\pi},$$

and $s = \text{sign}(x - \mu)$. We have $\widehat{R^{-1}}(x) \approx R^{-1}(x)$ using (23).

### G.6. Optimisation

All of the experiments use the Adam optimiser. Table 5 gives learning rate and batch size values.

Most experiments use pytorch defaults for other tuning choices. One exception is density estimation for CLIMDEX, where we follow the optimisation setup from Laszkiewicz et al. (2022). Another is the $\nu = 0.5$ variational inference example, where we clip the gradient norm to 5, which was beneficial to the stability of mTAF and TTFfix.

Table 5: Optimisation Hyperparameters.

| Experiment Group | Learning Rate | Batch Size |
|---|---|---|
| Synthetic Density Estimation (Section 4.1) | 5e-3 | None (full pass) |
| Real Data (Section 4.2) | 5e-4 | 512 |
| Variational Inference (Section 4.3) | 1e-3 | 100 |

## H. Variational Inference Diagnostics

We measure the quality of a density $q_x(x)$ produced by VI using two diagnostics based on importance sampling. This involves calculating $w_i = p(x_i)/q_x(x_i)$ where $p(x)$ is the target density and $x_i \sim q_x(x)$ (independently) for $i = 1, 2, \ldots, n$. In our examples we use $n = 10,000$.

Our first diagnostic is based on effective sample size (Robert & Casella, 2004)

$$ESS(n) = \left( \sum_{i=1}^{n} w_i \right)^2 / \sum_{i=1}^{n} w_i^2.$$

We report **ESS efficiency**, $ESS_e(n) = ESS(n)/n$. A value of $ESS_e(n) \approx 1$ indicates $q_x(x) \approx p(x)$.

Our second diagnostic is from Yao et al. (2018), who fit a GPD to the $w_i$s and return the estimated shape parameter $\hat{k}$. Lower $\hat{k}$ values indicate a better approximation of $p(x)$ by $q_x(x)$. Yao et al. (2018) argue that $\hat{k} < 0.7$ indicates a useful variational approximation, so we highlight this threshold in our tables.

## I. Data

Table 6 summarises the real data sets used in our experiments.

Table 6: Real data sets information.

| Name | Dimension | Average $\nu$ | Topic | Source |
|---|---|---|---|---|
| Insurance | 2 | 2.17 | Medical claims | Liang et al. (2022) |
| Fama 5 | 5 | 2.36 | Daily returns of 5 major indices | Liang et al. (2022) |
| S&P 500 | 300 | 4.78 | Daily returns of the 300 most traded US stocks | Novel to our paper |
| CLIMDEX | 412 | 4.24 | High dimensional meteorological data | Laszkiewicz et al. (2022) |

All datasets are pre-processed before density estimation is performed. To do so, data is normalized to zero mean and unit variance using the estimated mean and variance from the training and validation sets, as is standard practice in NF density estimation (Papamakarios et al., 2017; Durkan et al., 2019).

### I.1. S&P 500

Here we describe the S&P 500 dataset which we introduce. These financial returns are an example of moderately high dimensional multivariate data with extreme values. We take the closing prices of the top 300 most traded S&P 500 stocks, and convert them in standard fashion to log returns i.e. the log returns are $x_j = \log(\frac{s_{j+1}}{s_j})$ where $s_j$ is stock closing price on day $j$. We use data covering the time period 2010-01-04 to 2022-10-27, corresponding to 3227 days in total. In this example we concentrate on the tails of the data, rather than time series structure. As such, we treat each day of log returns as an independent observation in $\mathbb{R}^d$.

The test set is comprised of observations after 2017-09-14, with train and validation sampled uniformly from the period up to and including this date. This corresponds to 1292 training, 645 validation and 1290 test observations respectively.

# J. Additional Results for Density Estimation with Synthetic Data

## J.1. Other Experimental Settings

Table 7 expands on the results from Table 1 from the main paper, by including more values of $d$ and $\nu$.

Dashes in the table indicate potential unstable optimisation: at least one repeat had a very large final loss (above 1e5). In these cases, even excluding losses this large resulted in significantly worse mean losses than other methods. These results confirm that all the methods not specifically designed to permit GPD tails become increasingly unstable as the data becomes more heavy tailed.

The table includes results for the TTF_tBase architecture. As described in Section 4.1, this is similar to TTF, but uses a Student's T base distribution with trainable degrees of freedom. TTF_tBase always does worse than TTF and TTFfix. For the largest $\nu$ values – 2 and 30 – it is the worst of all methods. It also has the largest standard error values, suggesting that optimisation may be more difficult once these two methods are combined and there are two sets of tail parameters to tune.

Table 7: Density estimation results on synthetic example. Each entry is a mean value of negative test log likelihood per dimension across 10 repeated experiments, with the standard error in brackets. (Dividing by dimension acts to normalises the values, allowing for easier comparison across dimensions.) Bold indicates methods whose mean log likelihood differs from the best mean by less than 2 standard errors (of the best mean). A dash indicates potential unstable optimisation (at least one repeat had a final loss above 1e5). No entry indicates that the experiment was not ran.

| $d$ | Flow | $\nu = 0.5$ | $\nu = 1$ | $\nu = 2$ | $\nu = 30$ |
|---|---|---|---|---|---|
| 5 | normal | - | - | 2.01 (0.07) | 1.46 (0.00) |
| | m_normal | - | 403.24 (239.15) | 1.94 (0.02) | **1.46 (0.00)** |
| | g_normal | - | - | 1.93 (0.01) | **1.46 (0.00)** |
| | gTAF | 6.42 (0.27) | 2.49 (0.02) | 1.90 (0.01) | 1.46 (0.00) |
| | TTF | **3.33 (0.01)** | **2.34 (0.01)** | **1.89 (0.01)** | 1.47 (0.00) |
| | mTAF | 4.08 (0.03) | 2.49 (0.02) | 1.92 (0.01) | 1.46 (0.00) |
| | TTFfix | **3.33 (0.01)** | **2.35 (0.01)** | 1.89 (0.01) | 1.47 (0.00) |
| | COMET | 3.42 (0.01) | 2.35 (0.01) | **1.89 (0.00)** | 1.46 (0.00) |
| | TTF_tBase | 3.39 (0.02) | 2.51 (0.04) | 2.04 (0.02) | 1.65 (0.03) |
| 10 | normal | - | - | 2.00 (0.02) | 1.46 (0.00) |
| | m_normal | - | - | 2.04 (0.07) | 1.46 (0.00) |
| | g_normal | - | - | 1.98 (0.02) | **1.46 (0.00)** |
| | gTAF | 7.13 (0.31) | 2.63 (0.02) | 1.95 (0.00) | 1.47 (0.00) |
| | TTF | 3.55 (0.01) | 2.47 (0.00) | 1.93 (0.00) | 1.47 (0.00) |
| | mTAF | 4.48 (0.04) | 2.63 (0.01) | 1.95 (0.00) | 1.46 (0.00) |
| | TTFfix | **3.54 (0.01)** | **2.46 (0.00)** | **1.93 (0.00)** | 1.47 (0.00) |
| | COMET | 3.63 (0.01) | 2.46 (0.00) | **1.93 (0.00)** | 1.47 (0.00) |
| | TTF_tBase | 3.67 (0.01) | 2.63 (0.01) | 2.08 (0.01) | 1.62 (0.01) |
| 50 | normal | - | - | 2.02 (0.01) | 1.47 (0.00) |
| | m_normal | - | - | 2.02 (0.00) | **1.47 (0.00)** |
| | g_normal | - | - | 2.01 (0.00) | **1.47 (0.00)** |
| | gTAF | 7.49 (0.38) | 2.65 (0.01) | 1.99 (0.00) | 1.47 (0.00) |
| | TTF | **3.68 (0.00)** | **2.54 (0.00)** | 1.98 (0.00) | 1.47 (0.00) |
| | mTAF | 5.22 (0.04) | 2.62 (0.01) | 1.98 (0.00) | 1.47 (0.00) |
| | TTFfix | **3.68 (0.00)** | **2.54 (0.00)** | 1.98 (0.00) | 1.47 (0.00) |
| | COMET | 3.74 (0.00) | 2.55 (0.00) | **1.97 (0.00)** | 1.47 (0.00) |
| | TTF_tBase | 4.17 (0.01) | 2.84 (0.04) | 2.35 (0.04) | 1.82 (0.06) |

## J.2. Learned Tail Parameters

Figure 2 illustrates tail parameters learned by TTF corresponding to the $d = 5$ experiment of Table 7. These results show that the learned parameters don't exactly match the TTFfix values chosen to match the known tail shapes. Our theory – Theorem C.9 statement 2 – shows that the two should be equal for a good tail fit if our transformation were applied to

Final tail parameters for synthetic density estimation d=5

Figure 2: Box plots of TTF final tail shape parameters $\lambda$ from repeated optimiser runs. The dashed horizontal lines show the TTFfix values. The $y$-axis shows $1/\lambda$, as then the TTFfix values directly match the target Student's $T$ degrees of freedom.

a $\mathcal{N}(0,1)$ random variable. However Theorem C.9 statement 1 shows that changing the input random variable – e.g. to $\mathcal{N}(\mu, \sigma^2)$ – can produce a resulting distribution with a different tail shape. So the lack of a match in practice suggests the learned TTF parameters adjust slightly from the TTFfix values to compensate for the effect of the other normalising flow layers.

Adjusting in this way could be an advantage in learning the tail parameters (as in our TTF method) rather than fixing them in advance (as in our TTFfix method). However in this case any such advantage is small, as the overall results are very similar in Table 7. Also, as we have seen in other experiments, TTFfix is often competitive in practice, and can be easier to train.

