# OpenReview forum: "Flexible Tails for Normalizing Flows"
_ICML.cc/2025/Conference — ICML 2025 poster_

### Official Review · Reviewer_h4DD · 2025-03-10

**Overall Recommendation:** 4

**Summary:**

This paper addresses learning heavy tailed distributions with generative models and normalizing flows in particular. Similar to the previous approach COMET, it first transforms the tails of the input data to be light tailed and then applies a classical normalizing flow.

**Claims And Evidence:**

Let's span this by claimed contributions:

- "Illustrate the problem of using extreme inputs": I somewhat agree, but it is unclear how bad it is -- for the normal/m_normal/g_normal approaches normal, m_normal and g_normal, what is the success rate of training a model? Table 1 only says that for heavy tails, at least one out of ten runs fails.
- "We introduce ... TTF ... and prove it converts Gaussian tail to heavy tails with tunable tail weights" -- yes.
- "We demonstrate improved empirical results for density estimation ... and VI ... compared to standard NFs, and other NF methods for heavy tails." -- I think there are some important comparisons missing, see below.

# Update: Addressed

I am convinced that the method improves the quality of learning heavy tails with normalizing flows.

**Essential References Not Discussed:**

Seems like nothing is missed.

**Experimental Designs Or Analyses:**

In terms of density estimation, Table 2 merges several benchmarks from previous papers.

In terms of variational inference, it would be good to have a comparison to an established benchmark instead of a new artificial one.


# No update

**Methods And Evaluation Criteria:**

Datasets: Artificial targets make sense, cannot judge for the real-world data.

Metrics:

- KL: Correct me if I'm wrong, but the KL divergence = NLL is more sensitive to body than tail behavior due to the vanishing weight of these regions. Please also report the learned tails (which presumably even makes sense for the cases where the tails are learned and not fixed using the known values, given that there could still be some offset produced by the Lipschitz transformation).
- hat k: In the appendix, the authors write " Yao et al. (2018) argue that k>0.7 corresponds to a very poor approximation, and k<0.5 indicates a good fit.", but they then proceed to mark values already below 0.7 in Table 3 as bold, which is inconsistent. Also, more details on this metric would be crucial instead of the three lines of text in l. 349-351: How well do these metrics capture tails?

# Update: Addressed

Main argument: KL is indeed sensitive to extreme values, which I agree to.

**Other Comments Or Suggestions:**

One idea to strengthen the paper is by showing that a mixture of Gaussians with a finite number of components as a latent distribution does not have an influence on the tail behavior of the learned distribution. It seems like this is an easy proof -- the asymptotic behavior in any direction is determined by the largest standard deviation component only, and it is light tailed.

I read [Draxler et al. 2024] differently: From their Limitations section: "Secondly, it is unclear how the convergence metric [in] Section 5.2 is related to convergence in the loss used in practice, the KL divergence given in Equation (3)." So while I agree that they consider the full space (which contains the tails), their theory is not about KL convergence (they only have experimental evidence for KL). Given that their statements apply to any continuous distribution with finite second moment, it would be interesting to characterize what class of r is needed to convert any heavy into light tails.

l. 56: z should be x.


# Update: Found common ground

The authors will clarify the statements regarding universality, they might add the proof I suggested regarding the tails of GMMs being always light.

**Other Strengths And Weaknesses:**

No additional comments.

**Questions For Authors:**

What is the success rate of normal/g_normal/m_normal in Table 1?

How relevant is the Fréchet basin of attraction? What tails can/cannot be explained by it? Is this an argument for why TTF should be preferred over COMET?

Given that a heavy-to-light transformation should be applied on the data end: For sub-Gaussian tails, do the authors think that adapting the tails on the latent end is beneficial?

How well do NLL, ESS and k capture tails?

**Relation To Broader Scientific Literature:**

Learning heavy tails correctly is an important consideration for designing generative models, which can help them be successful in a broader set of applications. Like COMET, the method indicates that feeding heavy-tailed data into a neural network may be detrimental to performance; instead, the heavy tails should be converted first.

**Theoretical Claims:**

I did not check proofs, but the statements seem plausible.

The authors argue in the Related Work + Appendix B.2 that COMET is not in the Fréchet domain of attraction -- why is this important? COMET performs pretty well experimentally, and also I would assume the tail behavior can be rich, so why pick this particular instantiation? To my understanding, COMET could just well fulfill some other class of heavy-tailedness.

# Update: Largely addressed

The authors argue that COMET learns a class of less heavy-tailed distributions. I think this argument and the fact that they outperform COMET is good enough to motivate their method.

---

> ### Author Rebuttal · Authors · 2025-03-31
>
> Thanks for the detailed review which has provided some helpful new insights into our results.
>
> ## Success rate
>
> Failure rates for normal, m_normal, g_normal in Table 1 were 100% for $\nu=0.5$, and in the range 10-20% for $\nu=1$.
>
> For $\nu=1$, even in runs which did converge for these 3 methods, the best loss (NLL of 49.62) was significantly worse than results for other methods.
>
> These results confirm that all the methods not specifically designed to permit GPD tails become increasingly unstable as the data becomes more heavy tailed.
>
> (On checking our code, we note "did not converge" is not as we defined it in the paper. In fact it means at least one run had a final loss above 1e5, corresponding to extremely unstable optimisation. We'll update this in the paper.)
>
> ## NLL metric
>
> NLL has some sensitivity to both body and tail behaviour. For instance under a $N(0,1)$ density estimate, a single observed value of 10 would contribute a very large amount to NLL.
>
> We have concentrated on NLL as it's the main metric in the past literature on this topic. For instance Table 2 contains a direct comparison to log likelihood values from a previous paper.
>
> We agree it's desirable to explore metrics more specialised to tail behaviour in future e.g. see our reply to reviewer mY4v.
>
> ## Learned tails
>
> A rough plot of learned tail parameters for the experiment of Table 7 with $d=5$ is at [this link](https://pasteboard.co/oKRznj9WA3UL.png). The y-axis shows $1/\lambda$, with the target $\nu$ value shown as a horizontal line. The two would need to be equal for a good tail fit if our transformation were applied to a $N(0,I)$ random variable. As you suggest there is an offset due to the Lipschitz transformation.
>
> This is an interesting finding - thanks for suggesting this plot! It suggests a potential advantage to jointly learning the Lipschitz transform and the tail parameter. We'll write this up as an appendix.
>
> ## $\hat{k}$ threshold
>
> Table 3 highlights the threshold $\hat{k} < 0.7$ since Yao et al recommend this as a cut-off for a variational approximation to be useful.
>
> Thanks for pointing out that this isn't clear from our current discussion of Yao et al. We'll extend this.
>
> ## How well do VI metrics capture tails?
>
> ESS and $\hat{k}$ are general VI metrics. Informally we expect both are sensitive to tail behaviour. This is because both are based on importance weights $w(x) = p_{\text{target}}(x)/q_{\text{approx}}(x)$. When $p_\text{target}(x)$ is heavier tailed than $q_{\text{approx}}(x)$, this typically results in a few very large $w$ values leading to poor ESS and $\hat{k}$. We agree that specialised VI metrics for tail behaviour would be useful, but we are not aware of any in the related literature.
>
> ## Fréchet domain of attraction
>
> The Fréchet domain of attraction captures power-law (Pareto) tails and closely related tail behaviour. Our theory shows that COMET flows cannot represent such tails, although they can produce milder heavy tails similar to log-normal.
>
> This is an argument in support of TTF: it allows a much wider range of heavy tails than COMET flows.
>
> We'll add a sentence to summarise this in the main paper.
>
> ## VI benchmark
>
> We've chosen our VI example as it allows the tail weight to be varied. We found that established benchmarks didn't easily allow exploration of heavier tails, such as those corresponding to $\nu = 0.5$. We agree investigating established benchmarks is a useful future step.
>
> ## Mixture of Gaussians
>
> The suggested result should indeed strengthen the paper and be straightforward to prove, so we'll add it to an appendix.
>
> ## Draxler et al
>
> Thanks for pointing out that the theory of Draxler et al doesn't use KL convergence. Instead their metric is based on the effect of extending a sequence of coupling layers by one element. It's not obvious how to extend this to our setting where the sequence instead terminates in our TTF transform. So we'll remove our comments in D.4 on the possibility of extending the work of Draxler et al, and instead just note that it exists. This doesn't affect our main argument in Appendix D on extending universality under weak convergence.
>
> ## Line 56
>
> Thanks for highlighting this. We do mean z not x. Our presentation describes the "generative direction" of normalizing flows i.e. a function converting z to x. We do this simply to be concrete and consistent, and it does not affect our results. Note z can be extreme when a heavy tailed base distribution is used.
>
> We'll add a clarifying comment about "generative direction" around line 56 in the paper, as the current version doesn't mention this until line 139.
>
> ## Sub-Gaussian tails
>
> For heavy tails, we motivate making a transformation at the data end to avoid providing extreme input to a neural network. For sub-Gaussian tails this argument does not apply, so we expect that adapting the tail at either end will work well. Which is most beneficial likely depends on which is easier to implement in practice.

---

> > ### Comment · Reviewer_h4DD · 2025-04-02
> >
> > I thank the authors for their answers. I am happy with most answers, but ask the authors for some more comments below:
> >
> > > NLL metric
> >
> > I agree that a single "outlier" can have a significantly lower likelihood than body samples. I guess it is just a general problem for tail estimation that there is not a lot of data in the first place by construction.
> >
> > > Learned tails
> >
> > Thanks for the plot, but I am confused about what it shows! If I read it right, the predicted degrees of freedom is consistently off from the true value, right? How do you measure that exactly? Do you pick a direction and measure how the modeled log prob evolves?
> >
> > Similarly, can't one also use the modeled log prob to analyze the tail behavior also for the VI problems?
> >
> > > Fréchet domain of attraction
> >
> > So every distribution that COMET learns can be represented via TTF?
> >
> > > Draxler et al [and other universality work]
> >
> > Their argument goes from data to latent space, I think, so the TTF transform is applied in the beginning. This implies that their theory applies once the tails of the input distribution are finite (so that the variance is bounded and their assumptions are fulfilled). So this should provide a reasonable universal construction, although with the caveat of the the convergence metric.

---

> > > ### Author Response · Authors · 2025-04-08
> > >
> > > Thanks for the helpful comments.
> > >
> > > ## Learned tails
> > >
> > > I think we misinterpreted your original question slightly! The plot reflects our learned tail transformation **parameters** rather than the resulting tail **shapes**, which are hard to obtain. More explanation follows.
> > >
> > > "How do you measure that exactly? Do you pick a direction and measure how the modeled log prob evolves?"
> > >
> > > The plot shows the learned parameters for our tail transformation (after a transformation so they are comparable to degrees of freedom). The directions these parameters represent are aligned to each dimension in the original data. Since this example has the same tail shape for each dimension, we've simply aggregated all the parameters in each plot.
> > >
> > > "Similarly, can't one also use the modeled log prob to analyze the tail behavior also for the VI problems?"
> > >
> > > It would be very desirable to learn the tail shapes from (A) the fitted log prob in a density estimation context or (B) the log prob up to proportionality in a VI context. However it turns out this is not an easy problem. As we mention in our paper's conclusion, one possible approach for VI is "static analysis of probabilistic programs" (Liang et al 2024).
> > >
> > > "Thanks for the plot, but I am confused about what it shows! If I read it right, the predicted degrees of freedom is consistently off from the true value, right?"
> > >
> > > Yes the predicted degrees of freedom appear consistently off.
> > >
> > > This is because the plot shows the resulting degrees of freedom **if our transformation were applied to $N(0,1)$ tails**. However in reality the $N(0,1)$ base distribution is first transformed through some standard normalising flow layers. This means the tails are no longer $N(0,1)$ when our transformation is applied. Our theory shows this can alter the resulting tail shape. In particular Theorem B.9 shows that applying our transformation to $N(\mu, \sigma^2)$ tails produces a tail shape that depends on $\sigma^2$ as well as our tail transformation parameter.
> > >
> > > The plot is interesting because it confirms the optimal tail transformation parameters don't exactly match the known tail shapes. It appears they must be varied slightly due to the effect of the other normalising flow layers, as predicted by the theory. This suggests that ideally it's better to learn the tail parameters (as in our TTF method) rather than fixing them in advance (as in our TTFfix method). Despite this TTFfix is often competitive in practice, and can be easier to train. This is a helpful finding which we'll add to the paper's discussion.
> > >
> > > ## Fréchet domain of attraction
> > >
> > > "So every distribution that COMET learns can be represented via TTF?"
> > >
> > > This isn't covered by our theory. We prove that TTF can capture power-law like heavy tails (i.e. the Fréchet domain of attraction) but COMET cannot. The extreme value theory reviewed in our paper suggests this is the most important class of heavy tails.
> > >
> > > We haven't proved that TTF can exactly match the lighter tails generated by COMET (e.g. log-normal tails). To prove this we'd need to show that there's a suitable Lipschitz $T$ such that $R_\text{TTF} \circ T$ results in tails in the COMET class.
> > >
> > > ## Draxler et al
> > >
> > > Thanks for help on this point! The fact that their argument goes from data to latent space is very helpful.
> > >
> > > As you say, if our transformation does produce finite first and second moments, then we can indeed apply a form of the Draxler et al argument to show universality. This means we just need a small update to the appendix to correct its sketch argument.
> > >
> > > We may be able to provide a full proof for a camera ready version of the paper, but can't 100% commit to this as a rigorous asymptotic argument might have some complexities.

---

### Official Review · Reviewer_ocLN · 2025-03-13

**Overall Recommendation:** 4

**Summary:**

The paper introduces an invertible transformation to overcome difficulties that exist when fitting normalising flow models to heavy-tailed data distributions. The transformation is to be used at the tip of a normalising flow "chain", handling the heavy tails of the data and enabling normalising flows to be fit to heavy-tailed data without restricting the flow to a specific architecture.

## Update after rebuttal
No changes after the rebuttal, as no further comments were made by the authors and the interactions with other reviewers did not change my evaluation of the paper.

**Claims And Evidence:**

The claims made are supported by clear and convincing evidence.

**Essential References Not Discussed:**

I have not identified any lacking references.

**Experimental Designs Or Analyses:**

The experimental settings and analyses are valid, though I have not verified the contents in the Appendix.

**Methods And Evaluation Criteria:**

The evaluation criteria is sensible and uses reasonable baselines and data sets. There experimental settings cover both controlled scenarios and real data, allowing a clearer picture to be made.

**Other Comments Or Suggestions:**

No other suggestions or comments at this point.

**Other Strengths And Weaknesses:**

**Strengths**

The paper is very well-written, with an easy-to-follow structure. The method itself is quite simple to implement and understand, consisting of a simple transformation parameterised also in a straightforward manner, and is thoroughly studied by the authors. I also appreciate the authors discussed important limitations of their work.

**Weaknesses**

Given how much the authors rely on the theory and some of the notable aspects (such as universality) being so fundamental, I do wonder why the paper is organised with so much didactic introduction to already existing work. I do believe this point is rather subjective, so I consider this only a very minor weakness.

**Questions For Authors:**

No questions at this point.

**Relation To Broader Scientific Literature:**

The paper introduces a transformation to be applied to normalising flows to handle heavy-tailed data, thus relating closely to existing methods that address the same problem. Because it applies to most normalising flows that could be applied in tandem with the proposed method, it naturally relates to most of them.

**Theoretical Claims:**

The theoretical claims were verified insofar as covered by Appendix A.1, where I found no issues.

---

> ### Author Rebuttal · Authors · 2025-03-31
>
> Thanks for the positive review of our paper. We appreciate you taking the time to read it!

---

### Official Review · Reviewer_mY4v · 2025-03-15

**Overall Recommendation:** 2

**Summary:**

The paper proposes a way to enhance normalising flows models by incorporating ideas from Extreme Value Theory literature for representing distributions with heavy tails. The authors propose adding a new invertible layer after the traditional flow layers that does not have a Lipschitz transformation and hence provide more flexibility.

## update after rebuttal
I am still not convinced of the added value of the universality result and if the method can be applied to more classes of flow-based models, hence I will keep my score.

**Claims And Evidence:**

Yes, the evidence is convincing.

**Essential References Not Discussed:**

NA

**Experimental Designs Or Analyses:**

Yes!

**Methods And Evaluation Criteria:**

The evaluation follows commonly used  negative log-likelihood metric.

**Other Comments Or Suggestions:**

NA

**Other Strengths And Weaknesses:**

Taking motivation from Extreme Value Theory Literature was an ingenius idea

**Questions For Authors:**

I add most of my questions and concerns here.

1. Normalising flows are great models with attractive properties such as density estimation. Their real power appears to be in the form of Flow-Matching, though, where they have rivalled Diffusion models. The implicit modelling approach only requires learning the associated vector field instead of explicit mapping. Now my question is, if the problem of poor modelling of heavy tailed distribution also appears in flow matching. Since flow matching does not constrain the architecture, I doubt it. If it is still an issue, how could the proposed methods be applied to non-discrete versions of Normalising Flows?
2. If I understood correctly, the authors show that the additional layer does not change the existing universality proofs. In that case, I question the theoretical value of the TTF transformation, given that it doesn't relax any assumption on modelled class in the universality proof.
3.Whilst the density estimation results are better for the proposed method, I believe log-likelihood alone is not enough to show the power of the proposed additional layer, I believe metrics like MMD or TVD could shed more light on the efficacy of the method.
4. Could the authors show how the proposed layer help if the base distribution is chosen to be the T-distibution.

**Relation To Broader Scientific Literature:**

The prior works often consider using heavy tail distributions such as T-distributions (or Gaussian mixtures, generalised Gaussian), whereas the authors propose a more flexible approach and substantiate their claims empirically.

**Theoretical Claims:**

Yes, the theoretical claims appear valid.

---

> ### Author Rebuttal · Authors · 2025-03-31
>
> Thanks for the helpful review. We respond to each of your questions below, and invite you to increase your score if you think we have addressed the points fully.
>
> ## Flow-Matching
>
> Thanks for raising this interesting question: does it remain challenging to fit heavy tailed observations in Flow-Matching? We expect similar problems would occur here and in other related methods e.g. diffusion models and continuous normalizing flows. The reason is as follows.
>
> All these methods involve learning a vector field $u(x,t)$ controlling a differential equation. The vector field is taken to be a neural network. In a successful generative model we will sometimes have $x \approx x_i$ for any given training data item $x_i$. So if there are extreme values in the training data, the neural network will need to be able to cope with such input. A key argument of our paper is that neural networks are poor at this task.
>
> A plausible fix is to use our tail transform as a final discrete layer following a Flow-Matching generative model. Either joint training or a two-stage approach could be tried, similarly to the methods in our paper for discrete normalizing flows. Empirical investigation is beyond the scope of our current paper, but is a very interesting avenue for future work.
>
> We will add a paragraph to our discussion section commenting on this topic.
>
> ## Universality
>
> You ask a natural question about the theoretical value of the TTF transformation given that it does not improve existing universality properties. We do comment on this in the paper, but it may not be easy to find as it's in the appendices (Appendix D.3). Here's an extended version of our argument.
>
> A normalising flow defines a set of probability densities depending on the capacity of the flow (e.g. number of knots in a spline flow, number of neural network parameters etc). Roughly speaking, universality means that under the limit of large capacity the set contains all probability densities. We expect improved tail behaviour modelling means that it's easy to get good approximations from lower capacity. This is because only a small number of parameters are required to produce heavy tails. In contrast a spline flow with a Gaussian base distribution (for example) would need a large or infinite number of knots to modify a Gaussian tail to become heavier.
>
> In summary, universality properties are about the limit of infinite capacity. We expect the benefits of our approach occur for any given finite capacity. Further theoretical developments would be needed to formalise this argument mathematically, but it is consistent with our empirical findings.
>
> We'll add a brief summary of the above to the main paper's section on universality.
>
> ## Metrics
>
> Thanks for the suggestion of MMD and TVD as metrics. We have concentrated on test log likelihood as it's the main metric in the past literature on this topic. For instance Table 2 contains a direct comparison to log likelihood values from a previous paper. We've not been able to repeat our experiments to report additional metrics within the review period, but we will explore this in future work. We note care may be needed applying MMD in higher dimensional examples: https://arxiv.org/abs/1406.2083.
>
> ## Combining new layer and T-distribution for base
>
> Thanks for the intriguing suggestion of combining these approaches i.e. (1) a Students T base (2) our final TTF layer. In the paper we present these as two competing methods, and didn't consider making a combination.
>
> We have run some experiments on combining the methods. For the setting of Table 1 with $\nu=0.5$, we get a scaled NLL of 4.17 with standard error of 0.01. This is better than the heavy tailed base distribution methods, but not as good as the TTF methods.
>
> In another setting - $d=5$, $\nu=0.5$ - we find similar results. The combined method gets a scaled NLL of 3.40 with standard error of 0.02. Again this is better than the heavy tailed base distribution methods, but not as good as the TTF methods (see Table 7 in the appendices). However the differences between the methods are smaller now.
>
> Further experiments would be needed to explore why the combined method has worse performance.

---

> > ### Comment · Reviewer_mY4v · 2025-04-04
> >
> > Thanks for writing the rebuttal.
> >
> > Flow Matching: You wrote, "In a successful generative model ....... So if there are extreme values in the training data, the neural network....."
> > Please correct me if I am wrong here, your argument that this issue arises cos of the Lipschitz-constrained properties in the flow models, rather than any neural networks themselves. When learning the vector field, in my opinion, there are no constraints on the neural networks. The transformation map is realised by solving the ODE, which begs the question if there exists no vector field for which the ODE solution can exactly map between a heavy tail distribution and a Gaussian (or some other source).
> >
> > Universality: Your point elaborated on what I meant. As you put it, "We expect improved tail behaviour modelling means that it's easy to get good approximations from lower capacity.", this is empirical and some theory based on it would have been more valuable, instead of showing that the existing Universality arguments are still applicable.
> >
> > Indeed, most flow-based papers only report the likelihood (or maybe FID when images look prettier), but that does not mean we should rely on one evaluation metric (although we can have a primary metric, such as likelihood here. But I understand that the rebuttal period is usually hectic and it's not always possible to run all the experiments suggested by all the reviewers.
> >
> > Thanks for analysing the T-distribution combination idea. Indeed, the worst results were unexpected, I am curious as to why this has happened, but I won't push this point further as the combination is not what the paper discusses.

---

> > > ### Author Response · Authors · 2025-04-08
> > >
> > > Thanks for your comments, see below for our response. As before, we invite you to increase your score if you think we have addressed your points sufficiently.
> > >
> > > ## Flow Matching
> > >
> > > You write: "your argument that this issue arises cos of the Lipschitz-constrained properties in the flow models... The transformation map is realised by solving the ODE, which begs the question if there exists no vector field for which the ODE solution can exactly map between a heavy tail distribution and a Gaussian (or some other source)"
> > >
> > > This is indeed a crucial theoretical issue: under what conditions (e.g. Lipschitz) does a vector field exist which can produce heavy tails under flow matching or similar methods? However our response was about another issue: the difficulty of learning a vector field even when it does exist.
> > >
> > > Consider an extreme data point $x^*$ e.g. in scalar data $x^*=1000$. Consider a generative model based on a vector field $u(x,t)$. That is we solve an ODE $x'_t = u(x,t)$ over $t \in [0,1]$ where $x_0$ is sampled from $N(0,1)$. Suppose this vector field can generate $x^*$. Then for $t \approx 1$ the resulting ODE trajectory will have $x_t \approx x^*$. When solving the ODE we must calculate $u(x_t, t)$. So we must evaluate the neural network $u$ for extreme input $x_t$.
> > >
> > > In the paper we argue that training neural networks with extreme inputs is difficult. This motivates using a transformation - such as the one we propose - to transform the training data so that it no longer has heavy tails.
> > >
> > > ## Universality
> > >
> > > You ask about theoretical results on the benefits of more flexible tail modelling under finite capacity. We'd argue this is available from existing results in the literature and our paper. Here is a summary, which we'll put in Section 3.5 (Universality) of our paper:
> > >
> > > "It's been proved that some NFs have a universality property: 'the flow can learn any target density to any required
> > > precision given sufficient capacity and data' (Kobyzev et al., 2020). In Appendix D we show that many NF universality
> > > results are preserved when the TTF transformation is added as a final layer.
> > >
> > > As we've already seen, the situation under bounded capacity is different. Standard NFs cannot produce heavy tailed distributions (Jaini et al 2020, reviewed as Theorem 1.2 above). However adding our transformation does permit these (see Section 3.2).
> > >
> > > So theoretically our method improves the set of distributions which can be modeled under bounded capacity without sacrificing expressiveness in the limit of infinite capacity. The next section shows this is reflected by improved empirical performance modelling heavy tailed data."
> > >
> > > Your questions have been very helpful in developing this summary. The universality material is one of the most recent additions to the paper, and it's been useful to improve how we present and interpret it.

---

### Official Review · Reviewer_a6XW · 2025-03-23

**Overall Recommendation:** 4

**Summary:**

Normalizing Flows have been shown to be difficult to make work when data/density to be represented has heavy tails, something common to see to atleast some degree in tasks such as density estimation and Variational inference. Neural networks are known to have difficulty in converging during training where heavy tails are involved. Also some recent work has shown that existing standard NF algorithms with light tailed Gaussians as base distribution does not work well which serves as a motivation for this work. Some recent work has suggested to using heavy tailed distributions as base distributions, and then passing it through a Lipschitz transformation, this work suggests doing it the other way, using a standard light tailed ddensity as the base distribution while making the transformation non-Lipschitz so that the final transformation becomes heavy tailed. The transformation function to induce heavy tails has to be both invertible and differentiable for this to work, for which the authors emply the ecdf function
The experimental results show some improvements over the other algorithms based on the other approach on real world datasets but marginal improvements on synthetic datasets.

**Claims And Evidence:**

I think the paper has done a good job at supporting their claims with experiments. I particularly liked how sections in Appendix B3 and D are provided by authors to elaborate and explain the claims which they used as motivation in the paper. Section D supports previous claim made on universality by the work of Kobyzev et al. The appendix is so well done and covers the practical and finer optimization details quite adequately, which are especially needed to practitioners especially when it is common to encounter numerical overflow and underflow issues when working with heavy tailed and datasets containing outliers.

**Essential References Not Discussed:**

I am moderately familiar with the literature on flow methods and extreme value theory, I think the authors cover all the most noteworthy papers that I know of.

**Experimental Designs Or Analyses:**

I found the analysis adequate and supporting the claims made by the paper. I have also felt in my first hand experience that working with heavy tailed base distributions can be more tricky in VI annd NF especially in higher dimensions and challenging posterior, I have also felt that fixing the tail parameters results in more stable optimization than optimizing it along with other parameters, and this work also makes similar conclusions based on their experimental results.

**Methods And Evaluation Criteria:**

The authors used both real and simulated datasets in the experiments and had a proper cross validation set up to compare with the baselines.

**Other Comments Or Suggestions:**

1. I have personally seen left and right tails in literature in place of lower and upper, maybe you can mention that somewhere to make it clearer.
2. Can you find some real world dataset where you can show a bigger improvement in results in Table 2.
I would still like to accept this work as it is so well written and explained.

**Other Strengths And Weaknesses:**

1. I think the paper is a really nice read, it introduces the reader with a lot of interesting concepts and methods in EVT, Normalizing flow and tail-index estimation like modelling the tails and estimating shape parameter of the GPD to researchers who work with standard Normalzing flows or standard VI using light tailed Gaussians.
2. The appendix is well written and the paper makes an effort to elaborate whenever it touches a claim in the appendix.
3. The empirical results are mostly supportive, it is good to see that in some cases where the target is quite difficult, standard methods with heavy tailed base distributions can fail to converge ash shown in Table 1.
4. The paper lists its limitations in quite transparent and matter of fact way, something which is missing in a lot of papers.
5. For a user who is really concerned about accuracy, and maybe dealing with smaller datasets, the results are encouraging and Table 3 clearly shows that.
Weaknesses
1. I would have hoped for a bit better results, meaning seeing higher degree of improvement in Table 2.

**Questions For Authors:**

Listed above already.

**Relation To Broader Scientific Literature:**

The work uses many recently proposed algorithms as baselines which means that there is still interest in the community for this nature and direction of work.

**Theoretical Claims:**

I found the theoretical  claims well done and supoorted by arguments using theory from previous recent work esp. that of Jaini 20.

---

> ### Author Rebuttal · Authors · 2025-03-31
>
> Thanks for the positive review of our paper. We appreciate you taking the time to read it!
>
> ## Left/right tails
>
> This is a great suggestion. We will update the paper to include terminology around left/right tails.
>
> ## Additional experiments
>
> Unfortunately each experiment on real world data took several weeks to set up and run, including the time to find suitable datasets and perform preliminary data cleaning. So we aren't able to add any more datasets at this stage in the review process. We argue that our method does show reasonable improvement for real datasets. We summarise the argument below.
>
> Table 2 shows our method provides improvements for all experimental datasets. In line with our synthetic experiments these gains are modest for lower dimensional datasets (Insurance, Fama 5), but larger for higher dimensional datasets (CLIMDEX and S&P returns).
>
> In particular for the highest dimensional dataset, CLIMDEX, we reach a NLL value of -2214. This is a significant improvement on competing methods, which provided values of -2113 (gTAF), -2121 (mTAF), -2118 (COMET), and had only a modest advantage on the value of -2102 from using no tail method.

---

### Decision · Program_Chairs · 2025-05-01

**Decision:**

Accept (poster)

**Comment:**

Understanding how generative models handle extreme values and haevy tailed distribution is an important open task. This paper proposes a novel approach for handling haevy tails in normalizing and provide supporting theory.  The reviwers agreed that the paper is well written and that the prposed method and theory is intersting and novel (3 oout of 4 reviewers voted for accpetance apfter the rebuttal). Remaining questions where clearly answered during the discussion period.